# Counter-Current Learning: A Biologically Plausible Dual Network Approach for Deep Learning

**Chia-Hsiang Kao**
Cornell University
ck696@cornell.edu

**Bharath Hariharan**
Cornell University
bharathh@cs.cornell.edu

'

## Abstract

Despite its widespread use in neural networks, error backpropagation has faced criticism for its lack of biological plausibility, suffering from issues such as the backward locking problem and the weight transport problem. These limitations have motivated researchers to explore more biologically plausible learning algorithms that could potentially shed light on how biological neural systems adapt and learn. Inspired by the counter-current exchange mechanisms observed in biological systems, we propose counter-current learning (CCL), a biologically plausible framework for credit assignment in neural networks. This framework employs a feedforward network to process input data and a feedback network to process targets, with each network enhancing the other through anti-parallel signal propagation. By leveraging the more informative signals from the bottom layer of the feedback network to guide the updates of the top layer of the feedforward network and vice versa, CCL enables the simultaneous transformation of source inputs to target outputs and the dynamic mutual influence of these transformations. Experimental results on MNIST, FashionMNIST, CIFAR10, and CIFAR100 datasets using multi-layer perceptrons and convolutional neural networks demonstrate that CCL achieves comparable performance to other biologically plausible algorithms while offering a more biologically realistic learning mechanism. Furthermore, we showcase the applicability of our approach to an autoencoder task, underscoring its potential for unsupervised representation learning. Our work presents a direction for biologically inspired and plausible learning algorithms, offering an alternative mechanisms of learning and adaptation in neural networks. [1]

## 1   Introduction

In deep learning, *biological plausibility* refers to the properties that deep learning algorithms could respect to avoid inconsistency with current understandings of neural circuitry or violation of fundamental physical constraints, such as the localized nature of synaptic plasticity [Grossberg, 1987, Crick, 1989]. Consequently, error backpropagation (BP), despite its wide application, has been frequently criticized for its lack of biological plausibility, particularly for the following three challenges: (a) The *weight transport problem*, which arises because BP requires the feedback pathway to use the same set of weights as the feedforward process, a mechanism not observed in biological systems [Burbank and Kreiman, 2012, Bengio et al., 2015, Lillicrap et al., 2016]. (b) The *non-local credit assignment problem* arises because backpropagation relies on the global error signal to update the synaptic weights throughout the network, instead of depending on local errors derived from local loss

---

[1]Code available at https://github.com/IandRover/CCL-NeurIPS24

computation.[2]. (c) The *backward locking problem* occurs because, in BP, each data sample must await the completion of both forward and backward computations of the previous sample, impeding online learning capabilities [Jaderberg et al., 2017, Czarnecki et al., 2017]. These limitations have propelled the development of alternative credit assignment methods that aim to better align with biological principles and address these significant issues [Lillicrap et al., 2016, Crafton et al., 2019, Launay et al., 2020, Nøkland, 2016, Bengio, 2014, Lee et al., 2015, Ororbia and Mali, 2019, Meulemans et al., 2020, 2021, Dellaferrera and Kreiman, 2022, Shibuya et al., 2023].

**Reaching Biological Plausibility With a Dual Network Structure.** To address the weight transport problem, we leverage a dual network architecture for processing feedback signals, which uses a different set of weights from the forward network. To tackle the non-local credit assignment issue, we use pairwise local loss, computing the difference in layerwise activations between the feedforward and feedback networks, and ensuring the local loss only updates local weight parameters through gradient detaching. We approach the backward-locking problem by preventing the feedback network from reusing latent activations and output signals from the feedforward networks. Since the feedback network operates independently of the output (prediction), the forward and feedback processes can occur simultaneously. These enhancements make our approach not only more biologically plausible but also potentially more effective in complex scenarios.

**Analogy to Biological Counter-Current Mechanism.** Our scheme draws inspiration from nature's *counter-current exchange mechanisms*, observed in fish gills, animal vessels, and renal systems. These physiological mechanisms use an anti-parallel structure to optimize resource or energy exchange between two flows. Similarly, our dual network learning scheme allows the input signals in the forward network, flowing from input space to target space, to receive target domain information from the target-to-source signal flow (in the feedback network) and reciprocally share their source information. Therefore, we name our learning scheme "counter-current learning," as this reciprocal exchange mirrors the efficiency and optimization seen in biological systems.

**Contributions.** This paper aims to introduce counter-current learning as a novel, biologically plausible alternative. We validate our approach through experiments on MNIST, FashionMNIST, CIFAR10, and CIFAR100 using MLP or CNN architectures. Additionally, we demonstrate the effectiveness of our model in autoencoder-based tasks, which, to our knowledge, represents the first application of biologically plausible algorithms in this area. Our approach addresses key challenges, including weight transposition and non-local credit assignment, and partially mitigates the backward locking, providing a promising avenue for advancing biological plausibility.

## 2 Literature Review

**Target Propagation: Addressing Biological Plausibility.** The target propagation (TP) family [Bengio, 2014] and its variants (e.g., local target representations) [Ororbia et al., 2018, 2023], first explored in the late 1980s [Le Cun, 1986, Le Cun and Fogelman-Soulié, 1987], have been developed to optimize the neural networks by using locally generated error signals. TP explicitly constructs local targets for each layer using a separate feedback network. Take difference target propagation (DTP) Lee et al. [2015] for example, an idealized global target signal is computed based on the labels and the prediction error at the output layer. Then, the local idealized targets are generated by (1) propagating the idealized global targets through the feedback network and (2) computing a linear correction using the activations from the forward network. Subsequently, local losses are computed by comparing the layer activations with their corresponding local targets. The weights of both the forward and feedback networks are updated based on these local losses. Notable variants such as Direct Difference Target Propagation (DDTP) [Meulemans et al., 2020], Local-Difference Reconstruction Loss (L-DRL) [Ernoult et al., 2022], and Fixed-Weight Target Propagation Shibuya et al. [2023] further refine this approach by introducing mechanisms to improve feedback weight training and enhance the accuracy of local error signals. Despite these advancements, TP methods still encounter the backward locking issue, since TP methods depend on the forward network's outputs and intermediate activations to compute targets. Moreover, recent iterations of TP algorithms, such as DDTP and L-DRL, can be computationally expensive. They require additional feedback

---

[2]In biological systems, synaptic plasticity is believed to be governed by local learning rules, such as Hebbian learning, where synaptic changes depend on the correlated activity of the pre-and post-synaptic neurons [Dan and Poo, 2004, Bartunov et al., 2018, Whittington and Bogacz, 2019]

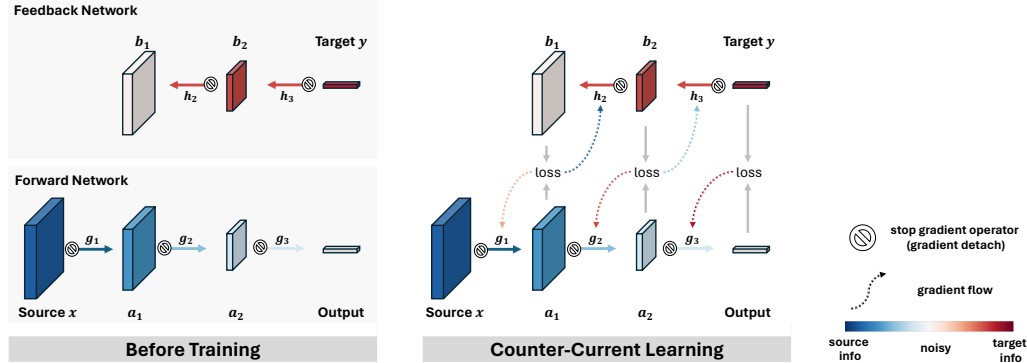

Figure 1: **Overview of the Counter-Current Learning Framework:** (a) **At initialization**, the counter-current learning framework establishes a dual network structure, with a forward network that maps the input to the target output, and a complementary feedback network that mirrors the forward network's architecture but propagates information in the opposite direction. The framework leverages the data processing inequality (DPI) from information theory, which states that information content cannot be increased through signal processing. Consequently, in both networks, information content decreases from the lower to the upper layers. (b) **During training**, the losses are computed in a layer-wise manner, i.e., by calculating the difference of activations from corresponding layer pairs between the forward and feedback networks, allowing the networks to learn from each other's complementary information. Notably, the dependency of the gradient on earlier layer parameters is interrupted using the gradient detachment operator.

weight update loop per data batch, leading to a three to six-fold increase in training time compared to traditional backpropagation [Meulemans et al., 2020, Ernoult et al., 2022, Shibuya et al., 2023].

**Other Efforts in Enhancing Biological Plausibility.** In addition to TP, several other methods have been proposed to overcome the biological implausibility of traditional backpropagation. The feedback alignment (FA) [Lillicrap et al., 2016, Nøkland, 2016, Crafton et al., 2019, Launay et al., 2020, Refinetti et al., 2021] family uses random feedback weights, instead of the transpose of the weight in the feedforward layer, to approximate the error gradient, thereby eliminating the need for precise synaptic symmetry. To resolve the backward locking problem, direct random target projection (DRTP) [Frenkel et al., 2021] proposed to randomly project the target signals to each layer as ideal targets. While achieving biological plausibility, DRTP encounters a significant performance drop concerning BP compared to FA algorithms Frenkel et al. [2021], Dellaferrera and Kreiman [2022]. Block-local learning (BLL) Kappel et al. [2023] explores block-wise target signal propagation; however, the algorithm requires backpropagation to update layers in the same block. Please refer to Section 6.5 for more literature review.

## 3   Counter-Current Learning Framework

In this section, we present the counter-current learning (CCL) framework, as shown in Figure 1, focusing on its formulation and key components.

**Setup and Feedforward Network.** Consider input space $\mathcal{X}$ and output space $\mathcal{Y}$, each with dimensions $d_0$ and $d_L$, respectively. The objective is to learn a mapping $F : \mathcal{X} \to \mathcal{Y}$ that minimizes the discrepancy between the predicted output and the target. We adopt an $L$-layered feed-forward neural network with activation function $\sigma$. Let $g_l(\cdot)$ denote the operation at layer $l$, and define $F_{fw} = g_L \circ g_{L-1} \circ \ldots \circ g_1$. Each $g_l$ is parameterized by weights $U_l$. The output of layer $l$ is $a_l = g_l(a_{l-1}) = \sigma(U_l a_{l-1})$, where $a_0 = x$.

**Feedback Network.** The proposed learning scheme introduces a complementary backward function $F_{fw}$ that mirrors $F_{fw}$ in an anti-parallel manner. $F_{bw}$ comprises layers $[h_L, \ldots, h_1]$, with each $h_l$ parameterized by weights $V_l$. We define $F_{bw} = h_1 \circ \ldots \circ h_L$. The output of layer $l$ is $b_{l-1} = h_l(b_l) = \sigma(V_l b_l)$, with $b_L = y$. The dimensions of hidden layers align between $F_{fw}$ and $F_{bw}$.

```python
from torch.nn import Linear, Module
import torch.nn.functional as F

class C2Model(Module):
    def __init__(self):
        super(C2Model, self).__init__()
        self.enc1 = C2Linear(784, 256)
        self.enc2 = C2Linear(256, 20)
        self.enc3 = C2Linear(20, 10)
    def fw_pass(self, x, detach):
        a1 = self.enc1.fw_pass(x, detach)
        a2 = self.enc2.fw_pass(a1, detach)
        a3 = self.enc3.fw_pass(a2, detach)
        return [x, a1, a2, a3]
    def bw_pass(self, target, detach):
        b2 = self.enc3.bw_pass(target, detach)
        b1 = self.enc2.bw_pass(b2, detach)
        b0 = self.enc1.bw_pass(b1, detach)
        return [b0, b1, b2, target]

class C2Linear(Module):
    def __init__(self, in_dims, out_dims):
        super(C2Linear, self).__init__()
        self.fw_layer = Linear(in_dims, out_dims)
        self.bw_layer = Linear(out_dims, in_dims)
    def fw_pass(self, x, detach):
        if detach: x = x.detach()
        return F.elu(self.fw_layer(x))
    def bw_pass(self, x, detach):
        if detach: x = x.detach()
        return F.elu(self.bw_layer(x))
```

```python
from torchvision.datasets import MNIST
from torch.utils.data import DataLoader
from nn.functional import one_hot
from torch import optim

def train_CCL_step(model, inputs, labels):
    fw_actvs = model.fw_pass(inputs, True)
    bw_actvs = model.bw_pass(labels, True)
    loss = 0
    for a, b in zip(fw_actvs, bw_actvs):
        loss += F.mse_loss(a, b)
    return loss

def train_BP_step(model, inputs, labels):
    fw_acts = model.fw_pass(inputs, False)
    return F.mse_loss(fw_acts[-1], labels)

train_dataset = MNIST(root='./data')
train_loader = DataLoader(train_dataset)
model = C2Model()
optimizer = optim.Adam(model.parameters())
for inputs, labels in dataloader:
    inputs = inputs.view(inputs.size(0), -1)
    # For CCL
    labels = one_hot(labels, 10).float()
    loss = train_CCL_step(model, inputs, labels)
    # For BackProp
    # loss = train_BP_step(model, inputs, labels)
    # For both CCL and BackProp
    loss.backward()
    optimizer.step()
```

Figure 2: **Code Snippet For Counter-Current Learning With Dual Network Architecture.**

**Stop Gradient Operation.** To address the backward locking problem and ensure local synaptic learning, we use the $\text{SG}()$ operation to decouple activations from weights in previous layers, disrupting the long error-backpropagation chain into local update segments. The $\text{SG}()$ can be implemented using PyTorch gradient detach operation easily. In CCL, each layer's input is processed with the $\text{SG}()$ operation. To avoid confusion, we use the hat symbol to denote the exact activations in the CCL paradigm. Specifically, for $1 \le l \le L$:

$$
\begin{aligned}
\hat{a}_l &= \hat{g}_l(\hat{a}_{l-1}) = \sigma(U_l \text{SG}(\hat{a}_{l-1})), \\
\hat{b}_{l-1} &= \hat{h}_l(\hat{b}_l) = \sigma(V_l \text{SG}(\hat{b}_l)),
\end{aligned}
\tag{1}
$$

where $\hat{a}_0 = x$ and $\hat{b}_L = y$.

**Loss Objective Function.** The objective of the counter-current learning algorithm is to minimize the difference between activations of $F_{fw}$ and $F_{bw}$ across all layers but the output layer. To avoid trivial solutions where activations converge to zero, we first reshape the activations into vectors and normalize them along the feature dimension using norm(). This normalization ensures meaningful alignment between forward and feedback passes :

$$
\min_\theta \sum_{l=0}^{L-1} \|\text{norm}(\hat{a}_l)\text{norm}(\hat{b}_l)^\top - I\|,
\tag{2}
$$

where $\theta = \{U_1, \ldots, U_{L-1}, V_1, \ldots, V_L\}$ are learnable parameters, and $I$ is the identity matrix. For the output layer ($l = L$), we use a different objective: the weight matrix $U_L$ is trained to minimize the cross-entropy loss between the network outputs $\hat{a}_L$ and the one-hot encoded labels $\hat{b}_L = y$.

**Biological Plausibility.** We examine the biological plausibility of the proposed counter current learning scheme. This framework mitigates the weight transport problem by using a different weight parameterization for the feedback network. For the non-local credit assignment problem, the update of the parameters is driven by local loss, instead of the back-propagated global error signals. Finally, we partially address the backward update problem by removing the dependency of the backward network and the forward network with careful gradient detachment.

**Implementation.** In Figure 2, we present a code snippet for the CCL algorithm in PyTorch, tailored for an MNIST classification. It shows the independence of the forward process and the feedback

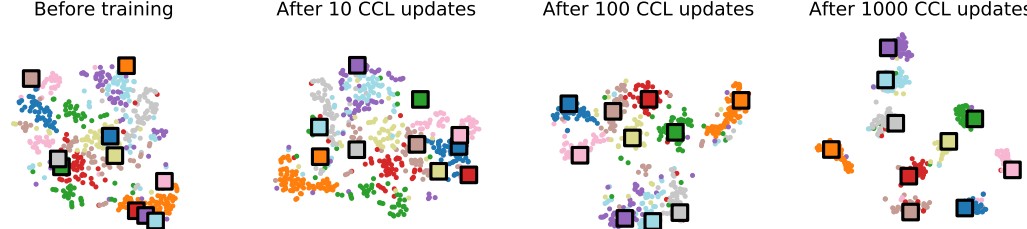

Figure 3: **Dynamic Feature Alignment Between Forward and Backward Models During Counter-Current Learning.** This series of t-SNE plots demonstrates the evolution of feature space alignment over different stages of training. Circular dots represent features from the forward network processing MNIST images, while squares depict features from the feedback network handling one-hot encoded labels. Each color represents a distinct class, with every subplot providing an independent t-SNE visualization. This emphasizes how distinct classes increasingly converge within and across the forward and backward models as training progresses, highlighting the dynamic and reciprocal nature of learning within the counter-current framework.

(backward) process from each other. The main `C2Model` module comprises three `C2Linear` layers, each inherits from `nn.Module` and consists of `fw_pass` and `bw_pass` for forward propagation and backward propagation, respectively. These functions accept an additional Boolean input, `detach`, allowing for the quick toggling between non-local (i.e., BP) and local learning (i.e., CCL) modes. For loss computation, the `train_CCL_step` function calculates the loss for counter-current learning. Conversely, the `train_BP_step` function computes the loss for error backpropagation.

## 4 Experiments

### 4.1 Counter Current Learning Facilitates Dual Network Feature Alignment

To investigate the alignment of latent features within the counter-current learning framework, we visualized embeddings from the penultimate layer of the forward network and the corresponding second layer of the feedback network at various stages of training. We employ a six-layer neural net trained on MNIST and analyze embeddings from both networks. t-SNE was applied independently to these embeddings at each training iteration to effectively visualize the evolution of feature spaces.

As illustrated in Figure 3, the embeddings from the forward model trained on MNIST data are represented by colored dots, while the embeddings from the backward model related to one-hot encoded labels are denoted by outlined squares of the same color. Throughout the training process, embeddings from the same class progressively align between the forward and backward models, suggesting that the forward and backward models mutually guide each other's feature representations toward a coherent and discriminative structure. Please refer to Appendix 6.3 for visualization of feature alignment of more layers and Appendix 6.4 for experiments on weight alignement between forward and backward layers.

### 4.2 Classification Performance

**Task Setup.** We evaluate the performance of our proposed method against several biologically plausible algorithms, including direct target propagation (DTP), DTP with difference reconstruction loss (DRL), local difference reconstruction loss (L-DRL), and fixed-weight difference target propagation (FW-DTP). The evaluation is conducted on MNIST, FashionMNIST, CIFAR-10, and CIFAR-100. All experiments are performed using stochastic gradient descent optimization with 100 epochs. We use cross-validation over 5 different random seeds and report the testing performance. The models are implemented using the PyTorch deep learning framework, and the code is available in the Supplementary Material. For the experiments on multi-layer perceptrons, we apply image normalization as a preprocessing step. For the convolutional neural network experiments, we use additional data augmentation techniques, including cropping and horizontal flipping. For the counter-current learning (CCL) algorithm, we search across different learning rates and gradient norm clipping values to find the optimal hyperparameters following cross-validation, as detailed in Appendix 6.1.

Table 1: Test performance on MNIST, FashionMNIST, CIFAR10, and CIFAR100, evaluated using multi-layer perceptrons. Performance metrics are reported for error backpropagation (BP), feedback alignment (FA), target propagation (DTP), DTP with difference reconstruction loss (DRL), local difference reconstruction loss (L-DRL), fixed-weight difference target propagation (FW-DTP), and cross-correlation loss (CCL). Best values per task are **bolded**, and second-best values are underlined.

|  | MNIST | FASHIONMNIST | CIFAR10 | CIFAR100 |
|---|---|---|---|---|
| BP [RUMELHART ET AL., 1986] | **98.19** $\pm$ 0.10 | **89.58** $\pm$ 0.25 | 50.03 $\pm$ 0.31 | **22.55** $\pm$ 0.19 |
| FA [LILLICRAP ET AL., 2016] | 96.96 $\pm$ 0.05 | 87.38 $\pm$ 0.12 | 45.76 $\pm$ 0.38 | 22.13 $\pm$ 0.41 |
| DFA NØKLAND [2016] | 97.27 $\pm$ 0.06 | 87.35 $\pm$ 0.99 | 42.86 $\pm$ 1.94 | 19.87 $\pm$ 1.50 |
| DRL MEULEMANS ET AL. [2020] | 93.05 $\pm$ 0.24 | 83.40 $\pm$ 0.18 | 42.09 $\pm$ 0.27 | 19.94 $\pm$ 0.28 |
| L-DRL ERNOULT ET AL. [2022] | 93.29 $\pm$ 0.21 | 83.60 $\pm$ 0.20 | 42.19 $\pm$ 0.30 | 19.96 $\pm$ 0.27 |
| FW-DTP SHIBUYA ET AL. [2023] | 97.20 $\pm$ 0.16 | 87.78 $\pm$ 0.47 | 45.91 $\pm$ 0.60 | 21.09 $\pm$ 0.31 |
| DRTP FRENKEL ET AL. [2021] | 92.16 $\pm$ 0.18 | 82.03 $\pm$ 0.56 | 33.85 $\pm$ 0.43 | 15.53 $\pm$ 0.33 |
| CCL (OURS) | 98.13 $\pm$ 0.10 | 88.58 $\pm$ 0.29 | **52.73** $\pm$ 0.59 | 21.76 $\pm$ 0.22 |

**Multi-Layer Perceptrons (MLP).** We follow Shibuya et al. [2023][3] for experimental setup and hyperparameter selection. For MNIST and FashionMNIST, we employ a fully connected network with 6 layers, each having 256 units. For CIFAR-10 and CIFAR-100, we use a fully connected network with 4 layers, each containing 1,024 units. We use the hyperbolic tangent activation function for BP and TP variant algorithms, while CCL adopts an ELU activation function. The results for MLPs are shown in Table 1, which demonstrates that the CCL obtains comparable results to error backpropagation and bears consistency with other biologically plausible algorithms.

Table 2: **Computational Complexity Comparison on MNIST and CIFAR10.** The computational efficiency of various learning algorithms is evaluated by measuring the estimated floating-point operations (FLOPs) per sample for a single training cycle. All measurements were conducted with a batch size of 32 samples. Values are reported in millions (M) of FLOPs, with the **best** performers highlighted for each dataset.

| ARCHITECTURE | MNIST 6 LAYERS | CIFAR10 4 LAYERS |
|---|---|---|
| BP | 2.39 M | 25.23 M |
| DTP | 48.69 M | 485.59 M |
| DRL | 125.73 M | 833.81 M |
| L-DRL | 82.45 M | 770.36 M |
| FWDTP-BN | 18.06 M | 170.08 M |
| CCL (OURS) | **2.55 M** | **27.89 M** |

Table 3: **Test Accuracy on CIFAR10 and CIFAR100 Using Convolutional Neural Network.** The metrics are reported for error backpropagation (BP) and cross-correlation loss (CCL).

|  | CIFAR10 | CIFAR100 |
|---|---|---|
| BP | 87.12 $\pm$1.76 | 51.92 $\pm$0.48 |
| CCL (OURS) | 82.94 $\pm$0.53 | 56.29 $\pm$0.25 |

**MLP Runtime Analysis.** We analyze the computational efficiency of various learning algorithms by measuring floating-point operations (FLOPs) per sample. The FLOPs estimation is performed using *torch.profiler* on the complete training cycle: forward (and feedback) propagation, loss computation, and backward propagation. Backpropagation (BP) serves as the baseline for comparison. Feedback Alignment (FA) and Direct Random Target Propagation (DRTP) are excluded from the analysis due to their algorithmic similarities with BP. All measurements were conducted with a batch size of 32

---

[3]Codebase: `https://github.com/TatsukichiShibuya/Fixed-Weight-Difference-Target-Propagation/tree/main`

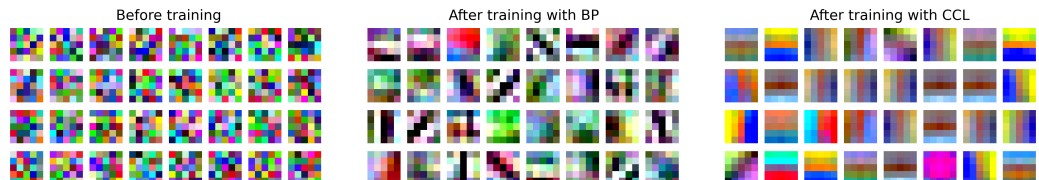

Figure 4: **Visualization of the First Layer Convolutional Kernels of the Forward Model Trained with Error Backpropagation (BP) and Counter-Current Learning (CCL).** Kernels from models trained with BP have more high-frequency components, as manifested as neighboring white (e.g., weight with high values) and black pixels (e.g., weight with low values). In comparison, those with CCL have more low-frequency components. We posit this might be because the error signal can contain more high-frequency information than the ideal target signal.

samples. The results in Table 2 show that CCL consistently outperforms the TP family algorithms in terms of training time. Note that the forward and feedback process for CCL is performed sequentially in this experiment.

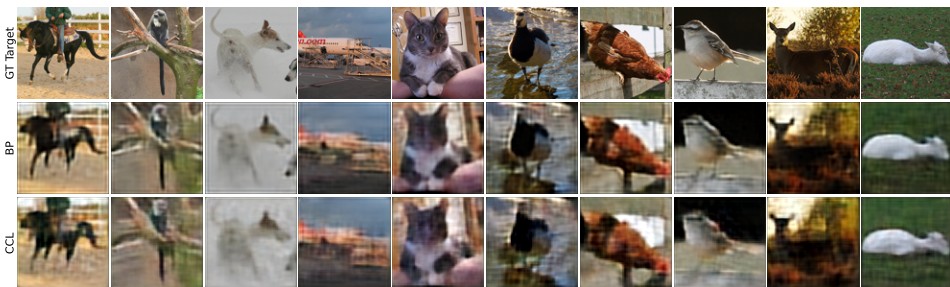

Figure 5: **Qualitative Comparison of an Eight-Layered Convolutional Autoencoder Trained Using Error Backpropagation (BP) and Counter-Current Learning (CCL).** The network structure does not contain skip connections. Testing set reconstruction results highlight CCL's comparable reconstruction as BP while achieving biological plausibility.

**Convolutional Neural Network.** We also evaluate CCL on convolutional neural networks (CNN) consisting of five convolutional layers (each with a kernel size of 3 and a max-pooling operation with a kernel size of 2) followed by a linear classifier, tested on CIFAR-10 and CIFAR-100. As shown in Table 3, our CCL-based model performs comparably to, or slightly inferiorly than, the BP-based model. Additionally, we visualize the kernels in the first convolutional layer to inspect the learned representations in Figure 4, demonstrating that CCL enables the model to learn meaningful convolutional kernels without using error backpropagation. Furthermore, we compare our CNN results on CIFAR-10 with those of L-DRL Ernoult et al. [2022] in Appendix 6.2, while FW-DTP Shibuya et al. [2023] does not include CNN implementations.

### 4.3 Auto-Encoder Performance

We explore the applicability of the counter-current learning (CCL) algorithm to autoencoders.

**Auto-Encoder on STL-10 Dataset.** A convolutional autoencoder with a four-layer encoder and a four-layer decoder is used. Different from the classification tasks, this architecture replaces the 2x2 kernel max-pooling with convolution layers with a stride of 2. Batch normalization is applied following each linear projection and before activation functions to ensure both stability and optimal performance. The hidden layers of the network are structured with dimensions of $[128, 256, 1024, 2048]$. Orthogonal weight initialization is used for training stability. Data augmentation techniques such as random cropping and horizontal flipping are incorporated. Hyperparameters including gradient clipping, learning rate, momentum, and weight decay are subjected to grid search, while cross-validation across five different seeds is employed to assess the reconstruction L2 loss.

**Results.** The test set's reconstruction metric—mean square error—is quantified as $0.0059 \pm 0.0001$ for BP and $0.0086 \pm 0.0001$ for CCL. The outcomes, illustrated in Figure 5, underscore the models'

proficiency on the test set. While both BP and CCL adeptly capture the general image structure, occasional artifact introduction, such as blurring, is observed in CCL compared to BP. This suggests that while CCL augments certain facets of autoencoder training, further refinement or architectural adjustments may be imperative to minimize visual artifacts and enhance detail preservation.

## 4.4 Empirical Analysis of Learning Dynamics for Counter-Current Learning

In this section, we provide insights into the functioning of the proposed Counter-Current Learning (CCL) algorithm by examining the representation similarity between the forward and feedback networks. We start by analyzing the feature similarity between randomly initialized feedforward and feedback models. Following this, we focus on the feature alignment in the high-level feature regime. Our investigation reveals the emergence of a reciprocal learning structure, where the top layers of both networks benefit from the bottom layers of each other during training.

We trained a model on the MNIST classification task using a configuration of five convolutional layers topped with a linear classification head. The training was conducted over 160 steps with a batch size of 32, achieving an average testing accuracy of $88.88\%$. To measure the cross-network representational similarity, we utilized Centered Kernel Alignment (CKA) Kornblith et al. [2019], a metric known for its robustness to invertible linear transformations. This was applied to evaluate layer and architecture similarity. The results, obtained using five different seeds, are presented as averaged CKA values on the test set. Figure 6 displays the horizontal axis marking the activations of forward layers, ranging from the input MNIST images to the logits (i.e., $a_6$), whereas the vertical axis denotes the activations of the feedback network starting from one-hot labels (i.e., target).

**Observation 1: Initialization Shows Noisy Top Layers and Misalignment Between Networks.** At initialization, our premise that the bottom layers contain more relevant information is validated. The initial CKA ($t = 0$, top-left subplot in Figure 6) reveals low similarity between the $a_6$ column (i.e., the top layer of the forward network) and the feedback network. Similarly, the $b_0$ row (i.e., the top layer of the feedback network) shows low CKA values with the forward layers, confirming our hypothesis of feature misalignment at random initialization.

**Observation 2: Alignment of High-Level Features During Training.** As training progresses, high-level features (i.e., the top layers of the forward network and the bottom layers of the feedback network) begin to show increasing CKA values (Figure 6, $t = 20$ to $t = 160$, top row). This trend suggests that the forward and feedback networks gradually align their high-level representations.

**Observation 3: Emergence of a Counter-Current Reciprocal Structure.** The dynamics of the counter-current learning algorithm reveal significant changes in CKA, particularly at higher layers (illustrated in the bottom subplots of Figure 6). Notably, the increases are concentrated in the $a_4$, $a_5$, $a_6$ columns of the forward network and the $b_0$, $b_1$, $b_2$ rows of the feedback network. This pattern supports the counter-current intuition that top layers benefit from the more informative bottom layers, fostering a reciprocal learning structure that guides each network's optimization process using the most informative features available.

This empirical analysis underscores the hypothesis that the counter-current learning scheme effectively leverages complementary and reciprocal alignment of representations between the forward and feedback networks. By exploiting the informative features from the bottom layers of one network to refine the noisy features in the top layers of the other, the counter-current signal propagation algorithm achieves a biologically plausible and efficient learning mechanism.

## 4.5 Ablation on Asymmetric Learning Rates for Dual Network Optimization

We delve deeper into the effects of varying learning rates between forward and feedback networks in CCL. As illustrated in Figure 7, our experiments confirm that asymmetric learning rates in forward and feedback networks facilitate effective and robust learning. Particularly noteworthy is our observation that robust learning outcomes are achievable even when the feedback network operates under a fixed random setting—specifically, with a zero learning rate (refer to the bottom rows of the figure). This suggests that the random projection of target labels by the feedback network conveys meaningful target domain information, echoing findings from the DRTP [Frenkel et al., 2021]. Moreover, our experiments indicate superior performance compared to the DRTP approach (refer to Table 1), possibly hinting that a random, non-linear neural network projection (i.e., CCL with a feedback

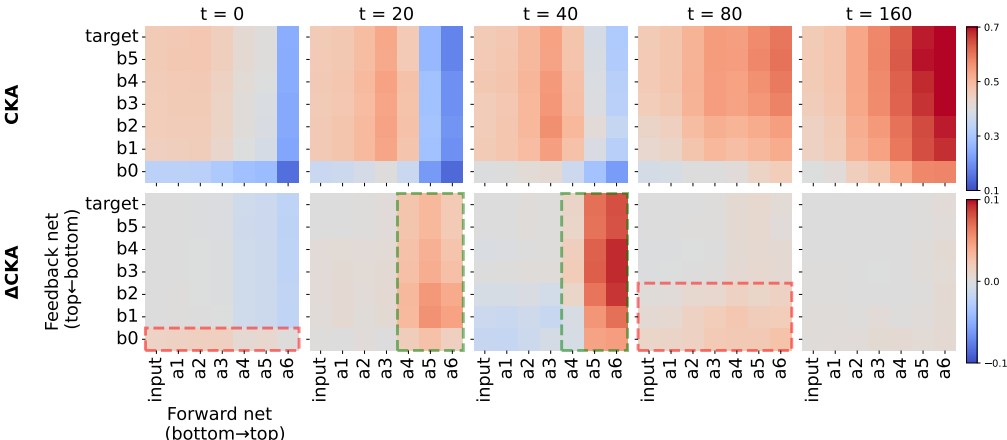

Figure 6: **Counter-Current Signal Propagation Enables Learning Through Reciprocal Representation Alignment. (Top) Centered Kernel Alignment (CKA)** between the forward and feedback networks during training. At the initial training step ($t = 0$), cross-network CKA is minimal, suggesting a low similarity between networks. As training progresses, CKA significantly increases, especially in the top layers of both networks, indicating high similarity in learned high-level representations. **(Bottom) Changes in CKA** between consecutive training steps (i.e., from step $t$ to step $t + 1$) reveal significant increases in the top layers of both networks, consistent with our counter-current learning insights. Notably, increases are concentrated in the $a_4$, $a_5$, $a_6$ columns of the forward network and in the $b_0$, $b_1$, $b_2$ rows of the feedback network, as highlighted by the green dotted box. These changes align with the expected reciprocal and complementary learning dynamics.

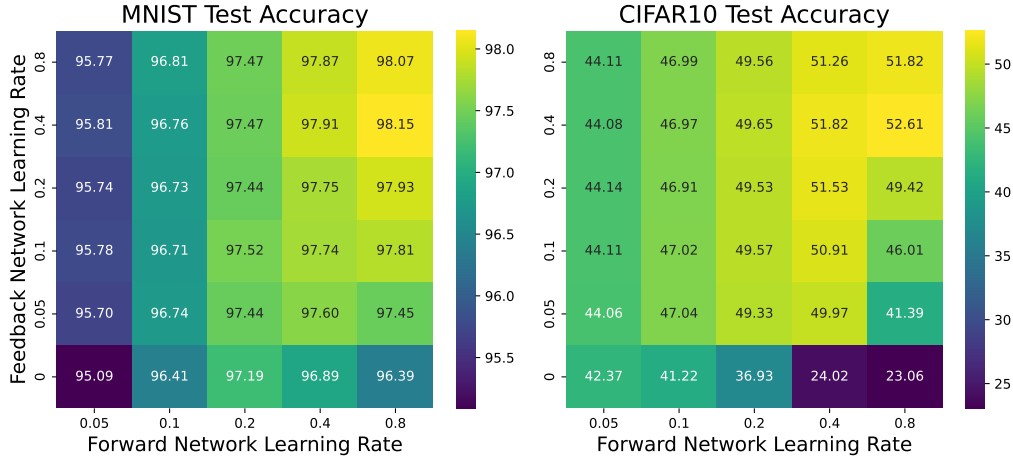

Figure 7: **Learning With Asymmetric Learning Rates in the Networks.** We investigate the influence of asymmetric learning rate in the forward and backward MLPs. This study demonstrates that effective learning can occur with asymmetric learning rates, even when the feedback network has a fixed random configuration (i.e., the learning rate for the feedback net is zero).

learning rate of zero) is more beneficial than a mere random linear projection (i.e., DRTP) of labels. This could be due to the neural network's ability to maintain and leverage more inductive biases, which can be crucial for the hierarchical learning processes.

## 5 Conclusion

In this paper, we introduced the counter-current learning framework, a novel approach addressing the critical limitations of traditional error backpropagation. Our dual network architecture enables a

dynamic and reciprocal interaction between feedforward and feedback pathways, supporting local learning and effectively resolving the backward locking problem. Our learning framework is validated across a diverse set of datasets including MNIST, FashionMNIST, CIFAR10, CIFAR100, and STL10, and in autoencoder tasks, demonstrates comparable performance with existing learning methods without compromising learning speed. This underscores the potential of our model to efficiently handle complex neural tasks and highlights its suitability for broader applications.

**Limitations and Future Directions.** We acknowledge several limitations that can guide future research in this field. Firstly, there is a need for further theoretical insights into the counter-current learning scheme, focusing on its learning dynamics, stability analysis, and inductive biases. Secondly, continuous exploration of this dual model architecture is essential, such as integrating residual connections or self-attention modules. Thirdly, hardware acceleration to streamline forward-feedback computation through parallelization could potentially reduce computation time and yield improved results within the same time frame. Lastly, exploring the integration of counter-current learning with other biologically plausible alternatives holds promise for advancing research in this domain.

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

# 6 Appendix

## 6.1 Experimental Setup

**Hyperparameter Search.** For experiments with MLP architectures, we conduct hyperparameter searches for each algorithm. We run all the combinations of the hyperparameters with 5 different random seeds and then select the hyperparameter set with the highest accuracies (or lowest loss for auto-encoder task) evaluated on the validation set and test on the testing set. For DTP, DRL, L-DRL, and FWDTP, we search across forward learning rates $[0.3, 1, 3]$, step sizes $[0.001, 0.003, 0.01, 0.03, 0.1]$, and backward learning rates $[0.0001, 0.0003, 0.001, 0.003, 0.01]$. Note that FWDTP-BN does not have the backward learning rate hyperparameter. For DRTP, the search space includes learning rates $[0.01 0.03 0.1 0.3]$ and mean ($[0.0 0.05]$) and standard deviation ($[0.01 0.03 0.1 0.3]$) for random project matrix. For BP and FA, we use learning rates $[0.4, 0.2, 0.1, 0.05, 0.02, 0.01]$ for hyperparameter search. and gradient clip values of $[0.5, 1]$. For CCL, the search space includes learning rates $[0.2, 0.5, 1, 1.5, 2]$ and gradient clip values of $[0.5, 1]$.

**Implementation Details.** Training MLP and CNN models with CCL can be unstable initially since both the forward and feedback networks are randomly initialized. We use learning rate warm-up for the initial 200 steps for CCL, which is adopted in Ernoult et al. [2022]. Moreover, unlike algorithms in the target propagation family [Meulemans et al., 2020, Ernoult et al., 2022], where the feedback network weights are trained using additional for-loops to tune the weights for each data batch, we introduce some techniques to stabilize the training course. We found that normalizing the activations during loss computation helps stabilize the training process. Additionally, as counter-current learning can also suffer from feature collapse, where a trivial solution to all pairwise losses is to produce constant activations, we introduce a remedy. Inspired by layer-wise training, for each latent activation $X \in \mathbb{R}^{b \times d}$, where $b$ stands for batch size and $d$ stands for feature dimension, we added an additional L2 loss to minimize the difference between $\text{norm}(X)\text{norm}(X)^\top$ and the identity matrix. Finally, in contrast to error backpropagation, where the error signals can be zero and the weight updates can be small at the end of training, the layer-wise losses in CCL are seldom zero, thus the weights can keep changing during the training. This can lead to worse minima. To accommodate this, we use gradient centralization [Yong et al., 2020] to centralize the gradient for parameters for both BP and CCL and flooding method [Ishida et al., 2020] to prevent some weights from updating if the sample-wise difference between output and target is smaller than 0.2 in CCL on CNN.

## 6.2 Comparison of Implementation

**Comparison with Ernoult et al. [2022].** We compare the results on CIFAR-10 using a VGG-like network architecture. As shown in Table 4, the results indicate that L-DRL [Ernoult et al., 2022] achieved similar testing accuracies to BP, reaching $89\%$. While these results are promising, it's worth noting some aspects of their methodology that may affect direct comparisons: (1) Their models were trained on the full training set, which differs from our approach; (2) The authors do not provide details on the hyperparameter search space or the implementation of cross-validation.

Table 4: **Comparison of Results on CIFAR10 Using VGG-like Convolutional Neural Network.** * indicates that we report the results from the literature. .

|  | OUR IMPLEMENTATION | L-DRL ERNOULT ET AL. [2022] |
|---|---|---|
| TRAIN ON VALIDATION SET | $\times$ | $\checkmark$ |
| BP | $87.12 \pm 1.76$ | $89.07 \pm 0.22^*$ |
| L-DRL ERNOULT ET AL. [2022] | - | $89.38 \pm 0.20^*$ |
| CCL (OURS) | $82.94 \pm 0.53$ | - |

**Comparison with Shibuya et al. [2023].** There are some quantitative differences in the results between ours and Shibuya et al. [2023]. Here, we identify the sources that can potentially lead to different empirical results:

- We note that the original implementation normalizes FMNIST, CIFAR10, and CIFAR100 using the mean and standard deviation of MNIST. We fix this by using each dataset's mean and standard deviation;
- We proceed with performing a hyperparameter grid search again using five random seeds because directly using the paper's best hyperparameters does not fit. We consider a slightly smaller hyperparameter search set than the original one, as shown in Section 6.1, due to the computation and time budget for the grid search of all the combinations.

## 6.3 Visualization of Representations in Other Layers

To show that the feature alignment can present in more than one layer, we show the results of a five-layered CNN model trained on CIFAR10. The model is trained for 3000 steps using CCL, and the t-SNE for layers 1, 3, and 5 with different time steps are shown in Figure 8.

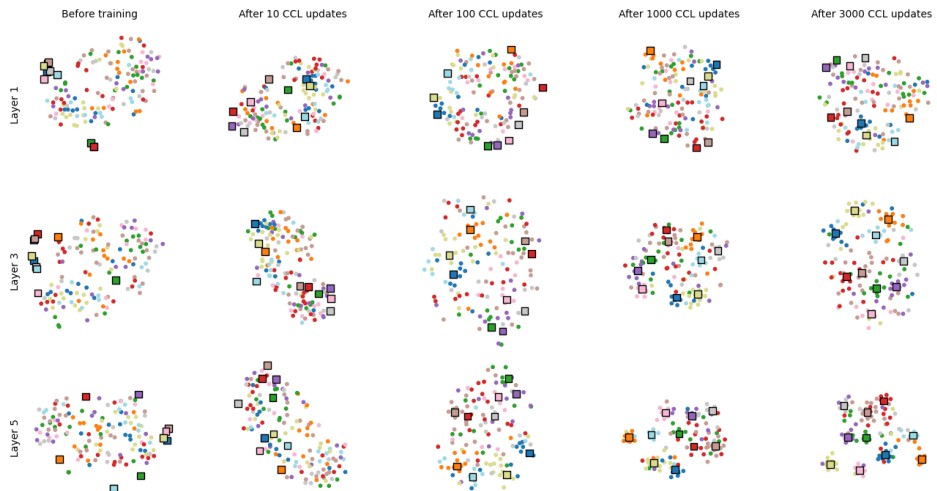

Figure 8: **Feature Alignments Across Layers.** We show that a similar feature alignment organization are observed in the third and fifth layer. While feature alignment in the first layer is inapparent.

## 6.4 Experiments on Weight Alignment

To better understand how CCL learns, we wonder if CCL learns by aligning the forward and backward weight. We compute the cosine similarity between the forward layer weights and the corresponding feedback layer weights for the model trained on MNIST using a five-layered MLP architecture.

As shown in Figure 9, our findings reveal two distinct phases during training:

- Phase one: Layers 1 and 5 show rapid alignment increases.
- Phase two: Alignments in layers 1 and 5 saturate. Alignments in the intermediate layers gradually increase until saturation.

Interestingly, intermediate layers showed a bottom-up convergence pattern, with layer 2 achieving highest similarity first, followed by layers 3 and 4. We found that high alignment isn't always achieved or necessary for effective learning. This may be due to (1) Dimensional reduction: Information propagation through the network affects alignment, and (2) Non-linearity: Activation functions impact the relationship between forward and backward weights. These factors may contribute to observed alignment patterns without requiring perfect symmetry between forward and backward weights.

## 6.5 Literature Reviews on More Biologically Plausible Algorithms

**Algorithms with only forward passes. Forward-forward algorithm (FFA)** [Hinton, 2022] only uses forward passes to update the neural network. Specifically, in the supervised learning scheme,

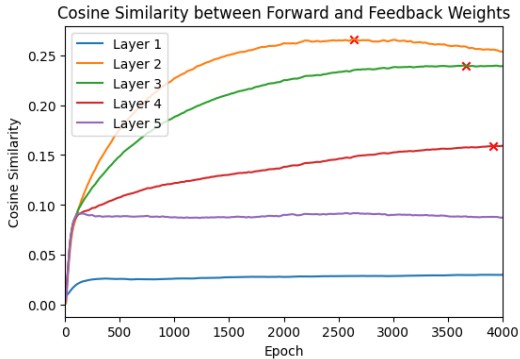

Figure 9: **Weight Alignment Between the Forward and the Backward Network Trained With CCL.** We find that the weight alignment of the first and the last quickly achieve plateau, while that of the intermediate layers increase gradually.

FFA sends the supervision signals with a recurrent architecture in a top-down manner, i.e., propagating the signal one layer by one layer for 3 - 5 steps. In contrast, our CCL scheme seeks to propagate the signal directly, thus attaining more computational efficiency. Similarly, **Error-driven Input Modulation (PEPITA)** [Dellaferrera and Kreiman, 2022] adopts two forward passes for updating the neural network, but the second forward pass requires a perturbed input, obtained via adding the input with the global error signals. Thus, PEPITA causes the update locking problem, and the neurons have to track two forward activities for loss computation, making it less biologically plausible. **SoftHebb** improves upon several state-of-the-art biologically plausible algorithms by achieving non-local property, mitigating time-locking of layer updates, and operating in an unsupervised manner. While SoftHebb and CCL perform similarly on CIFAR-10 and CIFAR-100 datasets, SoftHebb requires an extensive layer-wise hyperparameter search (24 hyperparameters). Additionally, SoftHebb's potential for auto-encoding tasks remains unexplored.

**Algorithms for spiking neurons.** BurstCNN [Greedy et al., 2022] and BurstProp [Payeur et al., 2021] propose to learn spiking neural networks using a two-compartment neuronal model. This model incorporates unique dendritic properties, synaptic plasticity, and high-frequency bursting, enabling information multiplexing across hierarchical levels. While these approaches offer high biological plausibility and attain good performance, extensions to auto-encoding tasks are lacking, and the required computation times are not provided.

