# OpenReview forum: "Counter-Current Learning: A Biologically Plausible Dual Network Approach for Deep Learning"
_NeurIPS.cc/2024/Conference — NeurIPS 2024 poster_

### Official Review · Reviewer_myCb · 2024-07-03

**Soundness:** 3
**Presentation:** 3
**Contribution:** 3
**Rating:** 6
**Confidence:** 4

**Summary:**

Inspired by the counter-current phenomenon observed in nature, the authors propose a counter-current learning (CCL) framework that decouples the input and backpropagation information in different circuits, overcoming many biological implausibility issues of backpropagation learning (BP).

**Strengths:**

I am not an expert in target propagation, but the general idea presented in this work sounds novel and interesting to me. I believe it also offers some insights into nature-inspired learning. I appreciate the constructive figures that help me quickly grasp the core idea of this work.

**Weaknesses:**

-	In Table 1, the authors should compare their results with other biologically plausible methods, like forward-forward and deep softhebb, or justify their criteria in model selection.
-	A minor drawback is the lack of validation in complicated dataset like Imagenet.

**Questions:**

-	I am surprised that the performance of BP is worse than the proposed local learning method. Can the authors explain more about this gap?
-	In forward-forward [1], Hinton proposes a recurrent network architecture that inputs the label at the top layer. This architecture seems similar to what is used in this work. Can the authors elaborate on the differences?


reference
[1]. Hinton G. The forward-forward algorithm: Some preliminary investigations[J]. arXiv preprint arXiv:2212.13345, 2022.

**Limitations:**

I think the authors have discussed it well.

---

> ### Author Rebuttal · Authors · 2024-08-06
>
> **[W1] Algorithm selection.**
>
> We appreciate the reviewer's question regarding concurrent work. A performance comparison is provided in general response 2 [GR2], demonstrating that our CCL algorithm outperforms other methods. While SoftHebb achieves similar results, it requires 24 hyperparameters compared to our 3, making our approach more efficient. We acknowledge the importance of comparing with other emerging biologically plausible approaches and will include more rigorous comparisons to methods like forward-forward and deep SoftHebb, as suggested.
>
> **[W2] More validation.**
>
> We appreciate the reviewer's questions regarding dataset and model scalability. However, due to the size and complexity of ImageNet, one typically requires batch normalization and other tricks in model architectures, which may not be biologically plausible. Additionally, recent work on biologically plausible learning algorithms (e.g., FA, DTP, DRL, FW-DT, and DRTP) has not evaluated these methods on large datasets like ImageNet, as datasets such as CIFAR-10 and CIFAR-100 demonstrate sufficient complexity for current studies in this field. We will include this discussion in the revised manuscript, acknowledging the need for further research on scalability to more complex datasets like ImageNet. Nonetheless, to showcase the capabilities of the CCL algorithm, we have applied it to an autoencoder task, which, to our knowledge, is the first time a biologically plausible algorithm has been scaled to an image autoencoding paradigm.
>
> **[Q1] Explanation for the result.**
>
> For a detailed discussion, please see General Response 3 [GR3] and Figure 1.a in the rebuttal document.
>
> **[Q2] Comparison to the forward-forward algorithm.**
>
> Quantitative comparisons are provided in the general response [GR2]. We discuss algorithmic differences here:
> - Compared to Section 3.3 in [4], CCL does not embed the label in the image. CCL adopts a feedback network to process and project the one-hot label, while in FFA the label is embedded in the input data, e.g., for MNIST training the first 10 pixels of an image is replaced with a one of N representation of the label.
> - Compared to Section 3.4 in [4], CCL does not need recurrent architecture to propagate the supervised signal backwardly: FFA adopts a recurrent network architecture to send the supervision signals in a top-down manner, i.e., propagate the signal one-layer by one-layer for 3 - 5 steps. In contrast, our CCL scheme seeks to propagate the signal directly, thus attaining more computational efficiency. Rolling out the signal transmission step-by-step would make the algorithm hard to scale to deep networks.
>
> [1] Lee, D. H., Zhang, S., Fischer, A., & Bengio, Y. (2015). Difference target propagation. ECML PKDD.
>
> [2] Meulemans, A., Carzaniga, F., Suykens, J., Sacramento, J., & Grewe, B. F. (2020). A theoretical framework for target propagation. NeurIPS.
>
> [3] Shibuya, T., Inoue, N., Kawakami, R., & Sato, I. (2023). Fixed-weight difference target propagation. AAAI.
>
> [4] Hinton, G. (2022). The forward-forward algorithm: Some preliminary investigations. arXiv preprint arXiv:2212.13345.
>
> [5] Nøkland, A. (2016). Direct feedback alignment provides learning in deep neural networks. NeurIPS.
>
> [6] Launay, J., Poli, I., Boniface, F., & Krzakala, F. (2020). Direct feedback alignment scales to modern deep learning tasks and architectures. NeurIPS.

---

> > ### Comment · Reviewer_myCb · 2024-08-10
> >
> > I appreciate the author's response, which addresses my main concerns. I think this is a promising bio-inspired learning method and will be interesting to a broad range of readers.

---

> > > ### Author Response · Authors · 2024-08-10
> > >
> > > We thank the reviewer once again for the constructive suggestion and the oportunity to clarify. We will include more related information in the future version. Have a great day!

---

### Official Review · Reviewer_kHhR · 2024-07-13

**Soundness:** 2
**Presentation:** 2
**Contribution:** 2
**Rating:** 4
**Confidence:** 4

**Summary:**

In this paper, the authors propose counter-current learning (CCL), a biologically plausible framework for credit assignment in neural networks.

**Strengths:**

The research direction of having new learning algorithms inspired biologically seems relevant and interesting.

**Weaknesses:**

- The authors discuss biological plausibility as a motivation, but do not discuss thoroughly how the proposed approach is biological plausibility. For instance, update-locking and non-frozen activities should be discussed in detail.

- The comparison with the state of the art in the domain is missing:
[1] Hinton, G. (2022). The Forward-Forward Algorithm: Some Preliminary Investigations. Technical report.
[2] Dellaferrera, G., Kreiman, G., and Kreiman, G. (2022). Error-driven Input Modulation: Solving
the Credit Assignment Problem without a Backward Pass. In Chaudhuri, K., Jegelka, S., Song, L.,
Szepesvari, C., Niu, G., and Sabato, S., editors, Proceedings of the 39th International Conference
on Machine Learning, pages 4937–4955. PMLR.
[3] Andreas Papachristodoulou, Christos Kyrkou, Stelios Timotheou, and Theocharis Theocharides.
Convolutional channel-wise competitive learning for the forward-forward algorithm. In Proceedings of the AAAI Conference on Artificial Intelligence, volume 38, pages 14536–14544, 2024.

- The results are sometimes counter intuitive. For example, the results of CCL vs BP for CIFAR100 in Table 3. Why is CCL working considerably better than BP?

- The presentation and organization of the paper can be improved. For instance, Figure 2 can be presented at a higher abstraction level as pseudo-code

**Questions:**

1. Could the author discuss the biological plausibility of their approach from weight transport, locality, freezing of neural activity, and update locking points of view?
2. Could the author provide some insight how their approach compares against or relates to the state of the art in the domain: FF, PEPITA, and CFSE?
[1] Hinton, G. (2022). The Forward-Forward Algorithm: Some Preliminary Investigations. Technical report.
[2] Dellaferrera, G., Kreiman, G., and Kreiman, G. (2022). Error-driven Input Modulation: Solving
the Credit Assignment Problem without a Backward Pass. In Chaudhuri, K., Jegelka, S., Song, L.,
Szepesvari, C., Niu, G., and Sabato, S., editors, Proceedings of the 39th International Conference
on Machine Learning, pages 4937–4955. PMLR.
[3] Andreas Papachristodoulou, Christos Kyrkou, Stelios Timotheou, and Theocharis Theocharides.
Convolutional channel-wise competitive learning for the forward-forward algorithm. In Proceedings of the AAAI Conference on Artificial Intelligence, volume 38, pages 14536–14544, 2024.
3. Why is CCL working considerably better than BP in Table 3 for CIFAR100?

**Limitations:**

The limitation of the work has been briefly discussed.

---

> ### Author Rebuttal · Authors · 2024-08-06
>
> **[Q1] In-depth discussion of bIological plausibility for counter-current learning**
>
> We thank the reviewer for the opportunity to further elaborate on the biological plausibility of our method from weight-transport, locality, freezing of neural activity, and update locking perspectives. Although the explanation has been provided in the manuscript Line 41-50 and Line 124-129 on how the model architecture and algorithm design mitigates these problems, we discuss more explicitly below
> - Weight-transport. We leverage a feedback network architecture for processing feedback signals, which uses a different parameter set from the forward network. The weights for feedback and forward networks are updated independently, mitigating the problem of weight transport by construction.
> - Locality. We leverage local layer-wise loss functions and gradient detach (i.e., stop gradient operation) to ensure that the parameters are updated by a local loss function, not a global error signal that propagates through the network.
> - Freezing of neural activity. The back propagation method requires keeping the neural activations for the global error back propagation since the process involves element-wise multiplication of the error signal with the derivative of the neural activations and thus it has to remain static. Our CCL algorithm, like all the algorithms in the Target Propagation Family, also requires freezing the neural activity for the loss computations. However, the duration of freezing time is halved with our CCL scheme since the two networks can process the signals simultaneously.
> - Update locking. In back-propagation, The latent activations of the forward network are kept and re-used for the backward phase.
> Moreover, the backward phase can only begin after the forward phase is finished, causing the update locking problem. In CCL, the backward process is performed through the feedback network, which is independent of the forward phase, and thus the forward and feedback processes occur simultaneously.
>
> **[Q2] Literature review and algorithm comparison**
>
> Quantitative comparisons are provided in the general response [GR2]. We discuss algorithmic differences here:
> - Forward-forward algorithm, FFA [1]
>   - Compared to Section 3.3 in [1], CCL does not embed the label in the image. CCL adopts a feedback network to process and project the one-hot label, while in FFA the label is embedded in the input data, e.g., for MNIST training the first 10 pixels of an image is replaced with a one of N representation of the label.
>   - Compared to Section 3.4 in [1], CCL does not need recurrent architecture to propagate the supervised signal backwardly: FFA adopts a recurrent network architecture to send the supervision signals in a top-down manner, i.e., propagate the signal one-layer by one-layer for 3 - 5 steps. In contrast, our CCL scheme seeks to propagate the signal directly, thus attaining more computational efficiency. Rolling out the signal transmission step-by-step would make the algorithm hard to scale to deep networks.
>
> - Error-driven Input Modulation, PEPITA [2]
>   - PEPITA requires two forward passes. In PEPITA, the parameter update function requires computing the difference between the two forward passes. This causes the update locking problem
>   - PEPITA requires the neurons to track two states. As PEPITA requires two forward passes,it requires every neuron to keep track of two neural activities, which is more biologically implausible.
>   - PEPITA does not follow locality. The global error signal is directly transformed and added to the latent feature; this does not follow the locality learning concept where synaptic changes depend on the correlated activity of the pre-and postsynaptic neurons.
> - Channel-wise Feature Separator and Extractor, CFSE [3]
>   - CFSE is a layer-wise training method that contains structural inductive bias. CFSE groups input channels in the forward stage, which can potentially provide inductive bias preferable for training.
>   - CFSE works with two different normalization methods. CFSE adopts batch normalization and group normalization methods to work, while CCL attains similar results without group normalization.
>   - CCL is validated on auto-encoder experiments, which CFSE does not.
>
> **[Q3] Explanation for the result on CIFAR-100.**
>
> For a detailed discussion, please see General Response 3 [GR3] and Figure 1.a in the rebuttal document.
>
> [1] Hinton, G. (2022). The Forward-Forward Algorithm: Some Preliminary Investigations. Technical report.
>
> [2] Dellaferrera, G., Kreiman, G., and Kreiman, G. (2022). Error-driven Input Modulation: Solving the Credit Assignment Problem without a Backward Pass. ICML.
>
> [3] Andreas Papachristodoulou, Christos Kyrkou, Stelios Timotheou, and Theocharis Theocharides. (2024). Convolutional channel-wise competitive learning for the forward-forward algorithm. AAAI.
>
> [4] Shaw, N. P., Jackson, T., & Orchard, J. (2020). Biological batch normalisation: How intrinsic plasticity improves learning in deep neural networks. Plos one, 15(9).

---

> ### Comment · Reviewer_kHhR · 2024-08-10
>
> I thank the authors for their response! I have gone through the manuscript and reviews multiple times.
>
> About the biological plausibility, I think there are still several very unclear areas. In particular, it is not clear how exactly the training is performed. I have also read the answer to other reviewers' questions, still I have trouble understanding the details and the details are where things become clear (e.g., is the definition of L_feature accurate?).
>
> 1. Weight-transport: The authors mention that "The weights for feedback and forward networks are updated independently, mitigating the problem of weight transport by construction." However, from the loss function you define (in Equation 2), is it possible that you end up with W^T (as in BP) for the backward pass? That is, you don't introduce the weight-transport explicitly, but implicitly the optimization is formulated in such a way that it ends up with the solution of BP. (The definition of the L_feature makes me wonder even more.) Also, what does this objective (minimize ||$\hat{a}_l-\hat{b}_l$||) mean really? Why should this optimization happen in the cortex or any biological system?
> 2. Locality: how do you optimize the loss function (in Equation 2)? It seems the loss function depends on all layers. From Figure 2, it seems that the optimization problem in Equation 2 is solved all together, isn't it?
> 3. Update locking: the update locking problem is defined in the literature (e.g., PEPITA): "Input signals cannot be processed in an online fashion, but each sample needs to wait for both the forward and backward computations to be completed for the previous sample." Isn't this also the case in CCL? In Figure 1/2 of CCL, it seems that although the forward and feedback passes can be simultaneously run, the first layer's update need to wait for the completion of the backward pass; the last layer's update need to wait for the complete of the forward pass.
> 4. Freezing of neural activity: I see that the authors acknowledge that CCL requires freezing the neural activity.
>
>
> One might argue that the whole motivation of this work is biological plausibility. If we want to sacrifice biological plausibility in any way, then we use BP, because it has generally the best performance.
>
> About the comparison with the state of the art, I remembered a new paper, how does your approach relate to this work:
> Ororbia, A.G. and Mali, A., 2019, July. Biologically motivated algorithms for propagating local target representations. In Proceedings of the aaai conference on artificial intelligence (Vol. 33, No. 01, pp. 4651-4658).
>
> About [GR3], could the author show that this is indeed due to overfitting of the BP or use common techniques to avoid overfitting?
>
> I'd appreciate it if the authors could clarify these point in their next response.

---

> > ### Author Response · Authors · 2024-08-11
> >
> > **[Q1] Clarification on biological plausibility**
> >
> > We sincerely thank the reviewer for the valuable time, insightful comments, and the opportunity to address ambiguities. We appreciate the chance to provide more detailed clarifications.
> >
> > **[Q1.1] On weight transport and locality**
> > - **Weight Transport**: We acknowledge the reviewer's concern about implicit weight alignment. In Figure 1.b (in rebuttal material), cosine similarity between the forward layer weights and the corresponding feedback layer weights are computed for the model trained on MNIST using a five-layered MLP architecture. Our analysis reveals a weak alignment between forward and feedback weights, with a maximum cosine similarity of 0.25 after training. This low similarity suggests that our method does not enforce weight transport; instead, even with weak alignment, the models attain good performance.
> > - **Loss Function**: The loss function (minimize ||$\hat{a}_l-\hat{b}_l$||) aims to align neural activities between corresponding layers in the feedforward and feedback pathways. This approach is inspired by local learning principles in neuroscience. It is related to Hebbian learning [1], where synaptic connections are strengthened when neurons on either side of the synapse have correlated activity such that the activity correlation would increase.
> > - **Locality**: Using Figure 1 (Counter-Current Learning subplot) as an example, one local loss function is ||$a_1$-$b_1$||. This means: (i) Function g1 is updated to make $a_1$ closer to $b_1$, and (ii) Function $h_2$ is updated to map $b_2$ closer to $a_1$, (iii) The stop gradient operation ensures locality by preventing ||$a_1$-$b_1$|| from influencing $h_3$, ||$a_2$-$b_2$|| from influencing $g_1$, and ||$Output$-$Target$|| from influencing $g_2$. This gradient detachment mechanism enforces the local nature of the updates.
> >
> > **[Q1.2] Update Locking and Activity Freezing**
> >
> > Our approach halves the time for activity freezing and partially mitigates the update locking problem.
> > To clarify our terminology:
> > - Backward locking problem: This is the term we use in our manuscript. It refers to the issue where the backward process must wait for the forward pass to complete [3, 4]. CCL allows simultaneous forward and backward passes, reducing this waiting time.
> > - Update locking problem: We acknowledge that while our method partially mitigates this issue, complete elimination remains a challenge for algorithms in Feedback Alignment Family, Target Propagation Family, and CCL.
> >
> > We will add more detailed discussions on weight transport, locality, update locking, backward locking, and activity freezing in the next version.
> >
> > **[Q2]  Whole motivation of this work is biological plausibility**
> >
> > We appreciate that the reviewer points out the trade-off between biological plausibility and performance. Indeed, biological plausibility is a primary motivation for our work, as it is for a growing body of research in the field, and the pursuit of biologically plausible learning algorithms can serve crucial purposes.
> > Our work, along with other approaches like Feedback Alignment, Target Propagation, PEPITA, and Forward-Forward Algorithm, aims to optimize neural network learning while addressing BP's biological implausibility. While we may not yet match BP's performance in all tasks, the insights gained from this line of research can advance our understanding of both artificial and biological neural networks.
> >
> > **[Q3] More literature review**
> >
> > The key differences between Local Representation Alignment (LRA) work by Ororbia and Mali (2019) and our approach are:
> > - Target generation: The lawyer-wise ideal targets in LRA require using global error signals (see Algorithm 1 in [2]), whereas CCL uses layer-wise comparisons between forward and backward activations.
> > - Backward locking: LRA doesn't resolve this issue; it still requires global error signals, meaning that the LRA backward process only begins after the forward process. CCL addresses it by allowing simultaneous forward and backward passes.
> >
> > We will review Local Representation Alignment (LRA) in our revised manuscript.
> >
> > [1] Hebb, D. O. (1949). The organization of behavior: A neuropsychological theory. Wiley.
> >
> > [2]  Ororbia, A.G. and Mali, A. (2019). Biologically motivated algorithms for propagating local target representation, AAAI.
> >
> > [3] Nøkland, A., & Eidnes, L. H. (2019). Training neural networks with local error signals. ICML.
> >
> > [4] Huo, Z., Gu, B., & Huang, H. (2018). Decoupled parallel backpropagation with convergence guarantee. ICML.

---

> > > ### Author Response · Authors · 2024-08-11
> > >
> > > **[Q4] Evidence of BP overfitting**
> > >
> > > As we cannot add additional plots in this discussion phase, we provide descriptive evidence of overfitting based on our experiments.
> > >
> > > Validation loss trend: For the BP model trained on CIFAR100 (Figure 1.a in rebuttal material), the validation loss reaches its minimum (cross-entropy loss of 2.23) at epoch 47. It then steadily increases to 2.99 by epoch 94, and settles around 3.01 at the end of training. We observe a growing gap between training and validation loss as training progresses, which is an indicator of overfitting.
> > >
> > > In our revised manuscript, we will include both training and validation loss plots. Thank you.

---

> ### Comment · Reviewer_kHhR · 2024-08-12
>
> I thank the authors for their response!
>
> > Weight Transport: We acknowledge the reviewer's concern about implicit weight alignment. In Figure 1.b (in rebuttal material), cosine similarity between the forward layer weights and the corresponding feedback layer weights are computed for the model trained on MNIST using a five-layered MLP architecture. Our analysis reveals a weak alignment between forward and feedback weights, with a maximum cosine similarity of 0.25 after training. This low similarity suggests that our method does not enforce weight transport; instead, even with weak alignment, the models attain good performance.
>
> I think the original issue was the extra biologically implausible assumption/constraint that the weights in the backward pass were the same as the forward pass. Now, while this is not the case here (at least not explicitly), with the objective function, it seems that the extra biologically implausible assumption/constraint is still introduced. One way to go around this issue is to say that this is actually the optimization problem that takes place in the cortex, which we discuss in the following.
>
> > Loss Function: The loss function (minimize $||\hat{a}_l -\hat{b}_l ||$) aims to align neural activities between corresponding layers in the feedforward and feedback pathways. This approach is inspired by local learning principles in neuroscience. It is related to Hebbian learning [1], where synaptic connections are strengthened when neurons on either side of the synapse have correlated activity such that the activity correlation would increase.
>
> I am familiar with the principle of "fire together, wire together." However, I don't see what is the reason for this cost function to be optimized by the cortex at one single-layer level locally. One could say, why should they fire together at all at one single-layer level? The objective to minimize $||\hat{a}_1 -\hat{b}_1 ||$ does not seem to be relevant at a higher level (e.g., for ultimately reducing the classification loss) for the cortex.
>
> > Locality: Using Figure 1 (Counter-Current Learning subplot) as an example, one local loss function is $||\hat{a}_l -\hat{b}_l ||$. This means: (i) ..., and (ii) ..., (iii) ...
>
> Do (i), (iii), (iii) happen in order in time? That is, first you do (i), then you do (ii), and finally (iii)? Or they are formulated as one joint optimization problem solved in python?
>
> > Update Locking and Activity Freezing: Our approach halves the time for activity freezing and partially mitigates the update locking problem.
>
> It seems there is some confusion about weight updates and passes. According to Nøkland2019: "The hidden layer weights cannot be updated before the forward and backward pass has completed. This backward locking prevents parallelization of the weight updates'',  and Huo2018 ''Backwards Locking – no module can be updated before all dependent modules have executed in both forwards mode and backwards mode''. In CCL, again, the first layer's update needs to wait for the completion of the backward pass; the last layer's update needs to wait for the completion of the forward pass.
>
> On the other hand, the authors mention that "Our approach halves the time for activity freezing". My question is: is that important? and, if so, why?
>
> Nøkland, A., & Eidnes, L. H. (2019). Training neural networks with local error signals. ICML.
>
> Huo, Z., Gu, B., & Huang, H. (2018). Decoupled parallel backpropagation with convergence guarantee. ICML
>
> I'd appreciate it if the authors could clarify these points in their next response.

---

> > ### Author Response · Authors · 2024-08-12
> >
> > We greatly thank your thoughtful review and the opportunity to clarify our work. While we believe CCL is at least as biologically plausible as other algorithms in the Target Propagation family (e.g., DTP, DRL, L-DRL, FW-DTP, LRA) compared to backpropagation, we appreciate this deeper discussion and stricter examination of biological plausibility.
> >
> > **[Q1]  Relevance of local objectives to higher-level cortical functions.**
> >
> > Thank you for this insightful question. We believe that a key contribution of other biologically plausible algorithms (e.g., Target Propagation Family) and CCL is demonstrating that solving local problems can lead to global loss reduction. These algorithms explore how such local computations might collectively contribute to higher-level functions, even without explicit global optimization. We'd welcome further discussion on this point if you have additional insights or concerns.
> >
> > **[Q2] Are local loss functions formulated as one joint optimization problem solved in Python**
> >
> > Yes, the local loss functions are formulated as a joint optimization problem solved simultaneously in our implementation.
> >
> > **[Q3] Confusion about weight updates and passes.**
> >
> > We appreciate the opportunity to clarify this point. We acknowledge that CCL does not fully mitigate the backward locking problem, as each module still needs to await both forward and feedback passes for updates. However, CCL partially addresses this issue by decoupling the feedback (backward) passes from the forward passes, distinguishing it from backpropagation and algorithms in the feedback alignment and target propagation families. While full mitigation might require layer-wise training methods, CCL represents an intermediate solution between no mitigation and full mitigation of the backward locking problem. We will revise our paper to more accurately state that CCL disrupts the dependency of the feedback/backward pass on the forward pass, rather than fully mitigating the backward locking problem.
> >
> > **[Q4] Importance of halving activity freezing time**
> >
> > By reducing activity freezing time, which is a byproduct of decoupling the feedback from the forward passes, CCL improves upon previous algorithms in feedback alignment and target propagation family and steps closer to how biological neural networks might operate by halving the latency. Moreover, CCL potentially offers computational advantages, particularly in resource-constrained scenarios. For instance, the reduced dependency between forward and feedback passes could open up possibilities for parallelization.
> >
> > We appreciate your critical analysis, which has helped us refine our presentation and clarify the contributions of CCL in the context of biologically plausible learning algorithms. We look forward to incorporating these insights into our revised paper.

---

> ### Comment · Reviewer_kHhR · 2024-08-12
>
> I thank the authors for their response and for bearing with my many questions!
>
> I try to summarize what we discussed below about biological plausibility:
>
> 1. Weight transport: As the authors mention, "We acknowledge the reviewer's concern about implicit weight alignment." They also provide nice insight that "CCL is demonstrating that solving local problems can lead to global loss reduction."
> 2. Locality: As the authors mention, "the local loss functions are formulated as a joint optimization problem solved simultaneously in our implementation." Therefore, I am not sure if the locality principle holds. Even if they stop gradient propagation the problem, in my understanding the problem is still not solved (entirely) locally.
> 3. Update locking: As the authors mention, they address this issue partially: "We acknowledge that while our method partially mitigates this issue..." and "We acknowledge that CCL does not fully mitigate the backward locking problem, as each module still needs to await both forward and feedback passes for updates."
> 4. Freezing activity: As the authors mention, they address this issue partially: "Our approach halves the time for activity freezing".
>
> Biological plausibility is one of the main motivations and in the title of the paper.
> As such, there is still several unclear areas around the biological plausibility of CCL. Therefore, as much as I think the paper has interesting contributions, my confidence in the biological plausibility of the approach is still low. Therefore, I slightly raise my score, since the discussion with the authors clarified a few points about the results and comparison with the state of the art, while the biological plausibility part remains largely unclear, at least to this reviewer.
>
> Once again, I thank the authors for their response and patience.

---

> > ### Author Response · Authors · 2024-08-13
> >
> > We appreciate the reviewer's thoughtful discussion on biological plausibility and concise summary of our exchange. We will incorporate these insights into the next version of the manuscript.

---

### Official Review · Reviewer_ijwx · 2024-07-13

**Soundness:** 3
**Presentation:** 3
**Contribution:** 3
**Rating:** 6
**Confidence:** 2

**Summary:**

In this work, the authors proposed the counter-current learning algorithm, a novel algorithm for biologically plausible training of feedforward neural networks. The learning rule is built upon the target-propagation algorithm and its variants. In these algorithms, the backward pathway is typically trained in a way that the target information propagated to each hidden layer provides a layer-wise target that nudges the forward pathway's output toward the true label. Instead, the authors jointly trained both feedforward and feedback weights projected to each layer, in a way that forward and backward pathways exhibit roughly the same activity at each layer. The authors applied this algorithm to various image recognition tasks (MNIST/Fashion-MNIST/CIFAR10/CIFAR100) and an autoencoder task and demonstrated that the proposed algorithm outperforms other variants of target-prop algorithms. The authors empirically analyzed the learning process of the algorithm by examining the development of hidden layer representations and through an ablation study.

**Strengths:**

The proposed learning algorithm is simple yet novel. The key novelty is the training algorithm for the backward pathway, where, unlike previous target-prop algorithms, activity matching between the forward and feedback pathway at each layer is used as the objective. I was positively surprised by the algorithm's strong performance because this learning algorithm clearly deviates from the backprop of the global loss. This training approach also makes the algorithm distinct from previous related approaches where the backward weights were fixed instead (Frenkel et al., Frontiers 2021; Shibuya et al., AAAI 2023).

**Weaknesses:**

The learning rules for hidden weights have trivial solutions, hence susceptible to representation collapse. The authors discussed their techniques for avoiding the collapse briefly in the supplement, but the implementation is not discussed in detail. Biological plausibility of the regularization mechanisms should also be discussed.

**Questions:**

Considering that the algorithm is minimizing the difference between $a_l$ and $b_l$, I expect to see large diagonal components in Figure 6 after learning. However, there is no diagonal trace in the figure. Why is that the case?

In addition, at t = 0, $b_1$ to target are already weakly aligned with the input, but $b_0$ is not aligned with the input. What is the origin of initial alignment and the absence of it in $b_0$?

On a related point, I wonder if the authors checked whether the stop gradient is implemented correctly, for instance, by calculating the gradients by hand and implementing them manually into the network.

**Limitations:**

The authors addressed the limitations, especially the lack of theoretical justification, nicely.

---

> ### Author Rebuttal · Authors · 2024-08-06
>
> **[W1] Discussion of biological plausibility of the regularization mechanisms.**
>
> We thank the reviewer for pointing out and allowing us to elaborate on this.
>
> - Normalization: Our use of activation normalization during loss computation aligns with divisive normalization observed in biological neural circuits [1]. This process helps maintain neural activity within a functional range, similar to the divisive normalization phenomenon observed in animals in the early 1990s in the primary visual cortex.
>
> - Modulation. The additional loss provides signals for surround modulation [2], a foundational property of visual neurons at many levels of the visual system (e.g., retina, visual cortex), auditory system, and so forth. Surround modulation describes that dissimilar stimuli between inside and outside neuron's receptive field would evoke stronger responses for a neuron, compared to similar stimuli. This is similar to the regularization methods used to prevent the model from representation collapse.
>
> - Flooding method: This approach, preventing weight updates when the sample-wise difference is small, can be likened to the concept of activation thresholds in biological neurons, where small input changes do not trigger action potentials.
>
> Implementation details:
> -  Modulation. We implemented a technique to prevent feature collapse for each latent activation $X ∈ R^{b×d}$, where $b$ represents the batch size and $d$ represents the feature dimension. We added an L2 loss term to minimize the difference between norm($X$)norm($X$)$^\top$ and the identity matrix. The loss is computed as: L_feature = $||norm(X)norm(X)⊤ - I||_F$
> - Avoid large gradient norms. We adopted gradient centralization [3] to mitigate the issue of large gradient norms. This technique centralizes the gradient vectors by removing their mean values before updating the model parameters: $g_{c} = g - mean(g)$, where $g$ is the original gradient and $g_c$ is the centralized gradient.
> - Focus on major differences. We implemented the flooding method [4] to focus the model on cases where predictions significantly differ from the ground truth. The modified loss function is expressed as $L_{flooded} = max(L, b)$, where $b$ is set to 0.2 as the original paper suggested.
>
> **[Q1] Why is there no diagonal trace in the figure 6.**
>
> The observed pattern in Figure 6 arises from the functional asymmetry when comparing forward and backward networks because their input signals are different. If we were to compare two independently trained forward networks using CCL, we would expect to see diagonal traces in the CCA analysis because both networks perform similar compression and learning functions. However, forward and backward networks exhibit functional asymmetry (e.g., one for feature extraction and another one for reconstruction), leading to rank disparity (e.g., the backward activations have a lower rank). This reduced detail and compressed information in the feedback activations result in lower pair-wise CCA values between forward and backward layers, manifesting as a non-diagonal trace in Figure 6.
>
> **[Q2] What is the origin of initial alignment?**
>
> Since MNIST is a relatively small and simple dataset, digits of the same class are relatively similar and are more likely to be projected to neighboring features in the latent space, forming several loose clusters. This would cause a relatively positive alignment in terms of CCA.
>
> **[Q3] Checking stop gradient implementation.**
>
> We followed your advice and checked the gradient implementation. We checked that the weight detachment is implemented correctly such that optimization over a local loss only involves the nearest layers.
>
> [1] Carandini, M., & Heeger, D. J. (2012). Normalization as a canonical neural computation. Nature reviews neuroscience.
>
> [2] Angelucci, A., Bijanzadeh, M., Nurminen, L., Federer, F., Merlin, S., & Bressloff, P. C. (2017). Circuits and mechanisms for surround modulation in visual cortex. Annual review of neuroscience.
>
> [3] Yong, H., Huang, J., Hua, X., & Zhang, L. (2020). Gradient centralization: A new optimization technique for deep neural networks. ECCV.
>
> [4] Ishida, T., Yamane, I., Sakai, T., Niu, G., & Sugiyama, M. (2022). Do We Need Zero Training Loss After Achieving Zero Training Error?. ICML.

---

> > ### Comment · Reviewer_ijwx · 2024-08-10
> >
> > Thank you so much for the revision and detailed responses.
> >
> > I have one comment regarding the new ELBO result.
> > The last equation appears to be incorrect. The final term should be:
> > $$ E_{q(z_1, z_2 | y)} \left[ \log \frac{q(z_1 | z_2)}{p(z_1 | x)}  \right] = E_{q(z_2 | y)} \left[ KL (q(z_1 | z_2) || p(z_1|x) ) \right], $$
> > since the KL divergence is evaluated with respect to $z_1$.
> > Additionally, the middle term, $E_{q(z_1, z_2 | y)} \left[ \log \frac{q(z_2 |y)}{p(z_2 | z_1)} \right]$, doesn’t seem to lend itself to a simple KL-based expression.
> >
> > Please let me know if I’m overlooking something.

---

> ### Author Response · Authors · 2024-08-12
>
> **Response to Reviewer Comments on Theoretical Derivation**
>
> We appreciate the reviewer's insightful comments, which have led us to refine our theoretical derivation. Based on their feedback and further consideration, we have revised our approach to better align with the CCL algorithm.
>
> **Problem with Previous Derivation**
>
> In our previous derivation, we incorrectly represented the relationship between variables. For example, in the term $E_{q(z_1, z_2 | y)} \left[ \log \frac{q(z_2 |y)}{p(z_2 | z_1)} \right]$, the $z_1$ in $p(z_2 | z_1)$ depends on $x$, not $y$.
>
> **Key Changes**
> 1. We now explicitly account for the relationship between $x$ and $y$ in the dataset $D$.
> 2. We have revised the ELBO derivation to more accurately represent the CCL process.
> 3. We have introduced an important approximation in the derivation that maintains consistency with the CCL algorithm's implementation.
>
> **Updated Derivation**
>
> The revised ELBO derivation is as follows:
>
> $$E_{(x,y) \sim D} \log p(y|x) \geq E_{(x,y)\sim D, q(z1,z2|y)}[\log (p(y,z_1,z_2|x) / q(z1,z2|y))]$$
> $$= E_{(x,y) \sim D, q(z_1,z_2|y)}[\log p(y|z_2,z_1,x)] - E_{(x,y)\sim D, q(z_1, z_2 | y)}[\log \frac{q(z_2 |y)}{p(z_2 | z_1,x)}]  - E_{(x,y)\sim D, q(z_1,z_2 | y)}[\log \frac{q(z_1 | z_2, y)}{p(z_1 | x)}] $$
>
> **Term-by-Term Analysis**
> - The first term can be understood as $E_{(x,y) \sim D, z_1\sim p(\cdot|x), z_2\sim p(\cdot|z_1)} [\log p(y|z_2)]$. This term represents the reconstruction of $y$ given the latent variables $z_1$ and $z_2$ sampled from the forward model, not the inference model.
> - The first latent alignment term can be rewritten as $E_{(x,y)\sim D, z_1\sim p(\cdot|x)} [KL(q(z_2 |y) | p(z_2 | z_1))]$. This term encourages alignment between the distribution of $z_2$ inferred from $y$ and predicted from $z_1$ (which is sampled from $p(\cdot|x)$, not from $q(z_1|y)$).
> - Likewise, the second latent alignment term can be rewritten as $E_{(x,y)\sim D, q(z_2 | y)}[KL(q(z_1 | z_2) | p(z_1|x))]$. This term encourages alignment between the distribution of $z_1$ inferred from $z_2$ and predicted from $x$.
>
> Consequently, the ELBO is $$E_{(x,y) \sim D, z_1\sim p(\cdot|x), z_2\sim p(\cdot|z_1)} \log p(y|z_2)]  - E_{(x,y)\sim D, z_1\sim p(\cdot|x)} [KL(q(z_2 |y)||p(z_2 | z_1))]  - E_{(x,y)\sim D, q(z_2 | y)}[KL(q(z_1 | z_2) || p(z_1|x))]$$
>
> **Connection to CCL Implementation**
>
> In our practical implementation of CCL, we approximate the last two KL divergence terms using L2 losses between latent features from forward and feedback networks. This approximation captures the essence of aligning the forward and backward pathways while providing a computationally tractable solution.
>
> [1] Luo, C. (2022). Understanding diffusion models: A unified perspective. arXiv preprint arXiv:2208.11970.
>
> [2] Ho, J., Jain, A., & Abbeel, P. (2020). Denoising diffusion probabilistic models. NeurIPS.

---

> > ### Comment · Reviewer_ijwx · 2024-08-13
> >
> > Thank you for your response.
> > However, I still find the derivation of both the first and second terms of the ELBO to be unclear and problematic.
> >
> > In the term-by-term analysis, the expression $\int dz_1 dz_2 q(z_1, z_2 | y) \log p (y | z_2)$ is replaced with $\int dz_1 dz_2 p(z_1, z_2 | x) \log p (y | z_2)$. These two equations are not equivalent unless $q(z_1, z_2 | y) = p(z_1, z_2 | x)$. Similarly, in the second term, it seems that $q(z_1, z_2 | y)$ has been substituted with $q(z_2 | y) p(z_1 | x)$. If additional approximations or assumptions were made in the derivation, they should be explicitly stated. The explanation in the term-by-term analysis paragraph may aim to clarify the approximation, but it lacks clarity. Additionally, if significant approximations were introduced, I would recommend avoiding the use of the term ELBO to prevent confusion.
> >
> > Although I believe that theoretical justification is not essential for this type of work, my overall confidence in the evaluation has decreased, and I have adjusted my rating accordingly.

---

> > > ### Author Response · Authors · 2024-08-13
> > >
> > > We sincerely appreciate the reviewer's thorough examination of our derivation. We acknowledge that our presentation introduced several approximations that were not explicitly stated. To improve the manuscript, we will:
> > >
> > > 1. Explicitly state all assumptions and approximations made in the derivation.
> > > 2. Remove the term 'ELBO' given the approximations involved, to avoid any misunderstanding.
> > >
> > > We understand that these issues have affected confidence in our evaluation, and we take these concerns seriously. We thank the reviewer for bringing these important points to our attention.

---

### Official Review · Reviewer_EBaT · 2024-07-15

**Soundness:** 2
**Presentation:** 3
**Contribution:** 2
**Rating:** 6
**Confidence:** 3

**Summary:**

In their paper ‘Counter-Current Learning: A Biologically Plausible Dual Network Approach for Deep Learning’ the authors introduce their new ‘counter-current learning (CCL) framework’ which they use to train neural networks of multiple network classes (MLPs, CNNs, Autoencoders) using a learning mechanism which solely relies on local calculation of learning signals, without the need to fully propagate gradients through the network. Through this, authors establish a link the brain’s counter-current exchange mechanism. The authors also show an analysis how feature representations are shaped over the course of learning in their learning framework. Whereas many biologically-plausible learning mechanisms have been proposed throughout the last couple of year, the author’s work particularly stands out due to its generality / easy of being applied to various different network classes used in machine learning.

**Strengths:**

As mentioned in the summary, I think many biologically-plausible learning mechanisms have been proposed throughout the last years and the author’s framework stands out to me due to it apparently generality as it seems to easily be applied to network classes which are, from my point of view, not commonly considered in bio-inspired learning mechanism papers, like autoencoders. This generality could allow for the authors framework to have an impact beyond theoretical neuroscience but also influence the more engineering focused side (e.g. neuromorphic computing). The new learning framework is supported by some additional empirical investigations, giving an intuitive understanding of the learning dynamics within the network.

**Weaknesses:**

While authors mention a potential biological link, not much space is given to discussing specific biological implementations and authors do not seem to make specific predictions which would allow for empirical validation of their proposed mechanism in biology, to validate whether the mechanism is actually utilised in the brain. As such, I see this paper as being more focused on the engineering / computing solutions side, where it does a good job, mentioned above. Given this focus on the computing side, I would have hoped that authors would also consider training a model using a current SOTA architecture (i.e. Transformers or State Space Models) using their technique, to show whether their algorithm can also handle these cases.

It would be important to discuss how the proposed work relates to a recent line of work in which two streams of information are assumed to exist in the brain: burstprop (Payeur et al. Nature Neuroscience 2021) and burstCCN (Greedy et al. NeurIPS 2022). Moreover, this work seems to have links with Sacramento et al. 2018 NeurIPS, in that both require specialised feedback neurons that encode the error signals. Both Sacramento et al. and Greedy et al. use an interneuron that learns to match the feedback, which conceptually seems to match the idea proposed by the authors. Perhaps even more relevant is the predictive coding of backprop by the group of Rafal Bogacz, which requires separate error neurons and both the forward and backward phases to align with each other. These should be discussed, and it may be worth making a more explicit link in future work.

**Questions:**

If the authors see their work primarily focused on an engineering solution, I am wondering whether their technique generalises to Transformers and / or State Space Models (like Mamba or Griffin). Otherwise, I wonder whether authors could make specific predictions based on the learning dynamics observed in their model which would allow the community to test for its plausibility using neural data.

How does your work relate to the bioplausible works referred to above?

**Limitations:**

I think authors address limitations on the machine learning / engineering side well but could potentially expand their discussion of limitations with regards to biological evidence for / against their proposed learning framework.

---

> ### Author Rebuttal · Authors · 2024-08-06
>
> **[Q1] Generalization of models**
>
> A: While MLPs and CNNs have established biological plausibility, the biological relevance of Transformers and State Space Models remains unclear and under-explored. For vision tasks, CNNs have been shown to match Transformer performance when computational resources, datasets, and techniques are equalized [1,2]. Recognizing the need for generalization, we've extended our research to auto-encoding tasks. To our knowledge, this represents the first application of a biologically plausible algorithm to an auto-encoder architecture, broadening the scope of our approach beyond traditional supervised learning paradigms.
>
> [1] Liu, Z., Mao, H., Wu, C. Y., Feichtenhofer, C., Darrell, T., & Xie, S. (2022). A convnet for the 2020s. CVPR.
>
> [2] Liu, S., Chen, T., Chen, X., Chen, X., Xiao, Q., Wu, B., ... & Wang, Z. (2023). More convnets in the 2020s: Scaling up kernels beyond 51x51 using sparsity. ICLR.
>
>
> **[Q2] Literature review and comparison**
>
> We appreciate the reviewer's suggestion to compare our Counter-current Learning (CCL) approach with recent work on dual-stream information processing in the brain. Here's a detailed comparison of CCL with BurstCNN/BurstProp, Sacramento, et al. (2018), and Predictive Coding:
>
> Network Structure:
> - CCL uses two separate networks: a feedforward network and a feedback network.
> - BurstCNN/BurstProp and Sacramento et al. (2018) use a single network with two firing modes.
> - Predictive Coding uses a hierarchical structure with prediction and error units at each layer.
>
> Error Representation:
> - CCL doesn't use global error signals; differences between forward and feedback local activations guide the learning.
> - BurstCNN/BurstProp encodes errors implicitly in burst firing patterns.
> - Sacramento, et al. (2018) allow local error signals to be generated by comparing different sources of neural input activity; the axonal output is a corrected target signal.
> - Predictive Coding uses explicit error neurons.
>
> Biological Plausibility:
> - CCL mitigates the update locking problem, allowing for more flexible and potentially more biologically plausible learning dynamics.
> - BurstCNN/BurstProp is based on observed burst firing patterns in neurons, enhancing biological plausibility. However, it still suffers from update locking problems.
> - Sacramento, et al. (2018) does not resolve the update locking problem, nor does it elaborate on multi-layered structures, limiting its biological plausibility in complex networks.
> - Predictive Coding aligns with theories of how the brain processes information, but explicit error encoding and propagation remains debated in neuroscience.
>
> Computational Complexity:
> - CCL allows for parallel processing in forward and feedback networks, potentially offering computational efficiency.
> - BurstCNN/BurstProp and Sacramento, et al. (2018) require distinct forward and backward phases, which may result in slower processing.
> - Predictive Coding involves iterative refinement, which can be computationally intensive.
>
> Our CCL approach offers several advantages in this context. By using separate forward and feedback networks, we avoid the update locking problem that affects BurstCNN/BurstProp and Sacramento et al. (2018). This separation allows for more flexible learning dynamics that could better reflect the parallel processing capabilities of biological neural networks. Moreover, CCL's approach to error representation - using differences between forward and feedback activations rather than explicit error signals - offers a unique perspective on how learning might occur in biological systems without the need for specialized error neurons or complex temporal coding schemes.
>
> We will add the above discussion over comparison in the camera-ready version.
>
> **[Q3] Testable prediction**
>
> We appreciate the reviewer's thought-provoking question. Based on our observations of the learning dynamics in our model, we propose the following testable predictions:
>
> - Organized increase in cross-layer neural activity correlation: As illustrated in Figure 6 of our manuscript, we observe an organized increase in cross-layer neuron activity correlation (highlighted by red and green boxes). This suggests a reciprocal and complementary learning dynamic among neurons. We predict that this phenomenon would be more pronounced when the network is exposed to novel, unseen patterns. This could be tested by measuring cross-layer neural correlations in biological networks before and after exposure to new stimuli.
>
> - Low similarity in local error signals across layers: Unlike backpropagation or related algorithms where error signals are globally coordinated, CCL's architecture suggests that error signals at different layers should exhibit low correlation or similarity. This prediction could be tested by comparing the similarity of local error signals across layers in biological neural networks, particularly during learning tasks.

---

> > ### Comment · Reviewer_EBaT · 2024-08-13
> > **Reply**
> >
> > Thank you we will improve our score.

---

### Official Review · Reviewer_sTDo · 2024-07-26

**Soundness:** 2
**Presentation:** 3
**Contribution:** 2
**Rating:** 6
**Confidence:** 4

**Summary:**

The article proposes a learning framework that addresses three major critiques of the backpropagation algorithm regarding its biological plausibility: (i) the weight transport problem, i.e., the error feedback weights being the transposes of the feedforward weights, (ii) the nonlocal update problem, i.e., the local weights being dependent on the global error signal, and (iii) the backward locking problem, i.e., the need to freeze layer activations until error feedback information becomes available.

The basic idea of the proposed framework is to construct a dual network that operates in the reverse direction to the original inference network. This dual-network has transpose dimensional weights (but not necessarily the transposes of the forward weights), with some similarity to the case of backpropagation. However, this dual network propagates the target value backward. Each layer of the dual network is compared to the corresponding layer (of the same dimension) in the original inference network. The loss is defined as the mean square error between the activations of the corresponding layers.  The numerical results demonstrate that the test performance in classification task are on par with backpropagation and the state-of-the-art models.

**Strengths:**

- Novelty: The article offers a novel approach for the biologically plausible learning framework. The use of a dual network to propagate the target value backward and using the mean square error between the corresponding representations of the network as the loss function appears to be a new idea.

- Numerical Experiments: The article presents several experiments demonstrating the performance of the proposed approach in comparison to existing methods. The performance appears to be close to that of backpropagation (and even significantly better in the case of CIFAR-10).

**Weaknesses:**

- Analytical Support/Discussion: There is no analytical justification provided for the proposed approach, making it unclear why it should work. This is the main weakness of the article. At the very least, there could be more discussion aimed at providing intuition about the underlying mechanism of the proposed approach that would enable effective training of neural networks. The authors cite "counter-current" mechanisms observed in nature as their primary motivation.

- Presentation: The presentation is generally smooth and clear. However, it can be further improved as addressed by the questions below.

- Reporting of Numerical Experiments: The results reported for the numerical experiments are confusing, especially as they conflict with those of Shibuya et al. [2023], whose experimental setup is adopted in the current article.

- Scalability: It is uncertain whether this idea would work for large-scale inputs/models (e.g., ImageNet 1K/ResNets).

**Questions:**

- How does the proposed approach train the forward neural network to minimize the mean square error (MSE) at the final layer?

- It would be useful to report the alignment (angle) between the forward and backward network weights, to check if the weight symmetry problem still exists.

- The sentence "Due to random weight initialization, the information content ... decreases ..." in the caption of Figure 1(a) is imprecise. How do you measure information content and relative to what, and why is it decreasing? Furthermore, the next sentence does not really apply to the "before training" (a) part of Figure 1.

- Figure 3: Do we see a similar organization of representations in other layers? Since the target label input for each class is fixed, we expect the backward network representations to be a single point for each class in any layer. Do the forward network layer activations appear as clouds around these points?

- Section 4.2: MLP Experiments: Why do the models used for comparison employ tanh nonlinearity, while CCL uses ELU? Test results with the same model properties would be more meaningful.

- Table 1: Is the DTP in the line below FA correct? As Noakland 16 is cited, is it actually DFA? Shibuya et al. [2023], which the authors claim to base their experiments on, report 52.17% accuracy for DTP/CIFAR10, which is not available or conflicts with Table 1. Similarly, Shibuya et al. [2023] report 51.33% for FA-CIFAR10, but Table 1 in the current article reports FA performance as 45.76%. The FW-DTP performances also do not match.

- Figure 4: It would make more sense to compare the conv. kernels of CCL after training with the corresponding kernels of BP, as the initial random kernels shown in this figure are not informative.

- Section 4.4: Do we need the CKA-based measure to compare the representations of the forward and backward networks? The proposed approach tries to minimize the mean square error (MSE) between them, so do we expect any linear transformation uncertainty? Isn't it more meaningful to look directly at the angle of alignment or the MSE normalized by the norm-square of the forward representations?

- Figure 7 and Section 4.5: The best performance appears to occur toward the top right corner, suggesting that feedforward-feedback learning rates should be similar (symmetric) rather than asymmetric?

**Limitations:**

The authors sufficiently discuss the limitations of their work at the end of the article. Related points are raised in the weaknesses and questions part.

---

> ### Author Rebuttal · Authors · 2024-08-06
>
> **[W1] We provide a theoretical extension for the existing framework**
>
> We provide an analytical framework for understanding CCL in General Response 1 [GR1]. In short, we show that CCL can be considered as optimizing an ELBO for a hierarchical model.
>
> **[W4] Validation on large-scale dataset/models**
>
> We acknowledge that the ImageNet dataset poses significant challenges due to its size and complexity. It typically requires batch normalization, whose biological plausibility is under-explored and often necessitates extensive architectural changes. Additionally, recent work on biologically plausible learning algorithms (e.g., FA, DTP, DRL, FW-DT, and DRTP) has not evaluated these methods on large datasets like ImageNet, as datasets such as CIFAR-10 and CIFAR-100 demonstrate sufficient complexity for current studies in this field. We will include this discussion in the revised manuscript, acknowledging the need for further research on scalability to more complex datasets like ImageNet.
> Nonetheless, to showcase the capabilities of the CCL algorithm, we have applied it to an autoencoder task, which, to our knowledge, is the first time a biologically plausible algorithm has been scaled to an image autoencoding paradigm.
>
> **[Q1] What is the loss for the final layer?**
>
> We thank the reviewer for reminding us of this. We adopt cross-entropy loss for the final layer, which is not explicitly mentioned in the paper. We will modify the content accordingly.
>
> **[Q2] Suggestion on reporting weight alignment**
>
> We appreciate this insightful suggestion. Please refer to General Response 4 [GR4] for discussion.
>
> **[Q3] Suggestion on caption for Figure 1(a)**
>
> We thank the reviewer for checking the details. The main idea is that signal processing cannot increase information, which corresponds to the data processing inequality (DPT) concept in information theory [1]. From a more intuitive perspective, due to dimensional reduction and the random initialization of the weight parameters, the information of the input signals can be lost during data processing. We will revise the sentence accordingly. Also, we agree that the sentence “Notably, … operator” should move to “(b) During training,” and we will also revise this accordingly.
>
> [1] Tishby, N., & Zaslavsky, N. (2015). Deep learning and the information bottleneck principle.
>
> **[Q4] Do we see a similar organization of representations in other layers?**
>
> Yes, similar organization is presented in other layers, especially for deeper layers. In Figure (2) in the rebuttal document, we show the results of a five-layered CNN model trained on CIFAR10. The model is trained for 3000 steps using CCL, and the t-SNE for layers 1, 3, and 5 with different time steps are shown.
>
> **[Q5] Why does CCL utilize ELU?**
>
> We find that ELU empirically gives better results than Tanh in CCL training when we perform exploratory analysis on MNIST in the early stage of research. We posit the main reason is that Tanh is a symmetric function and thus would induce feature collapse when trained with CCL. For example, at the beginning of training, say the first entries of $a_l$ (i.e., activation from forward network layer $l$) and $b_l$ (i.e., activation from feedback network layer $l$) are 1 and -1 respectively, then solving the local loss function, the weight would be updated such that the first entries of $a_l$ and $b_l$ becomes 0. This would lead to feature collapse for both the forward and feedback networks. We appreciate the reviewer for the insight and will add this to the Supplementary Section.
>
> **[W3, Q6] Result inconsistency from Shibuya, et al.**
>
> First of all, we thank the reviewer for noting the error, where the citation for DTP should be corrected. We will revise it in the camera-ready version.
>
> Second, we would like to kindly remind the reviewer that the statistics Shibuya et al. [2023] reported are the test error (%), not the test accuracy (%) shown in Table 1.
>
> Third, we would like to identify the sources that can potentially lead to different empirical results:
> - (a) We note that the original implementation for the code is incorrect, where the original codebase from Shibuya, et al. (2023) normalizes FMNIST, CIFAR10, and CIFAR100 using the mean and standard deviation of MNIST. We fix this by using each dataset’s mean and standard deviation;
> - (b) We proceed with performing a hyperparameter grid search again using five random seeds because directly using the paper’s best hyperparameters does not fit. We consider a slightly smaller hyperparameter search set than the original one, as shown in Section 6.1, due to the computation and time budget for the grid search of all the combinations.
>
> The above all can give rise to the difference between the results presented in our work and the original results.
>
> **[Q7] Suggestion on presenting BP and CCL convolutional kernels**
>
> We show the result in Figure (3) in the rebuttal document, where kernels learned by BP and CCL are visually different. Kernels from models trained with BP have more high frequency components, as manifested as neighboring white (e.g., weight with high values) and black pixels (e.g., weight with low values). In comparison, those with CCL have more low frequency components. We posit this might be because the error signal can contain more high-frequency information than the ideal target signal.
>
> **[Q8] Suggestion on similarity measurement**
>
> We thank the reviewer for pointing out this part. In Figure 6, we not only compare the similarities between pairwise features in the same layer (i.e., the diagonal entries) but also across different layers. We believe that using CKA can better capture relations across layers since it considers invertible linear transformation and orthogonal transformation.
>
> **[Q9] Does Figure 7 suggest feedforward-feedback learning rates should be symmetric?**
>
> Yes, it suggests that symmetric learning rates perform better.

---

> > ### Comment · Reviewer_sTDo · 2024-08-10
> >
> > I would like to thank the authors for their response. They have largely addressed my comments. I believe the approach presented in this paper has both merit and novelty, with the potential to impact future research on biologically plausible networks. Therefore, I have increased my overall rating.

---

> > > ### Author Response · Authors · 2024-08-10
> > >
> > > Thank you again for your constructive suggestions from the theoretical framework, additional qualitative experiments, discussion, and clarification. We appreciate your time in and we will incorporate them in the future version.

---

### Author Rebuttal · Authors · 2024-08-06

We thank the reviewers for their insightful feedback. We're grateful for their assessment of our work as novel (sTDo, ijwx, myCb), distinctive (ijwx), and empirically supported (sTDo, EBaT), with EBaT noting its 'apparent generality'. We address the main critiques below, focusing on theoretical foundation, literature review, result clarification, and intuition explanation. We will add the corresponding results and discussions to the next version of the paper.

**[GR1] A theoretical explanation and interpretation of CCL**

We appreciate reviewer sTDo for requesting a more analytical approach to explain and support the proposed counter-current learning (CCL) algorithm. We offer the following analysis through the lens of variational inference and hierarchical latent variable models.

CCL fundamentally aims to learn a conditional distribution p(y|x), where x represents input data and y represents target outputs. The algorithm employs a hierarchical structure, for example, with latent variables z1 and z2, utilizing both forward (encoding) and backward (decoding) pathways. The key insight is that CCL uses two different graphical models with which CCL can be interpreted as optimizing the Evidence Lower Bound (ELBO) on the log-likelihood of y given x:
- For the generative model p: x → z1 → z2 → y
- For the inference model q: y → z2 → z1 → x

### ELBO Derivation
Starting from the goal of maximizing log p(y), we derive the ELBO as follows with simplifications based on the conditional independence from graphical models:

$\log p(y|x) = \log E_{q(z1,z2|y)}[p(y,z1,z2|x) / q(z1,z2|y)]$

$≥E_{q(z1,z2|y)}[\log (p(y,z1,z2|x) / q(z1,z2|y))]$ (by Jensen's inequality)

$=E_{q(z1,z2|y)}[\log p(y|z2) + \log p(z2|z1) + \log p(z1|x) - \log q(z2|y) - \log q(z1|z2)]$

$=E_{q(z2|y)}[log p(y|z2)] - KL(q(z2|y) | p(z2|z1)) - KL(q(z1|z2) | p(z1|x))$

The CCL algorithm implements this variational framework in a simplified and computationally tractable form. The implemented losses connect to the ELBO terms as follows:
- Target reconstruction: This first term represents the reconstruction of y given the latent variable z2. In CCL, it is implemented as a loss between the output of the forward pathway and the target y.
- Latent consistency: The two KL divergence terms encourage alignment between the distributions of z2 (z1) inferred from y (z2) and predicted from z1 (x). In CCL, this is approximated by L2 losses between latent features from forward and feedback networks. While this simplification assumes zero variance in the distributions, it captures the essence of aligning the forward and backward pathways. Using Monte Carlo sampling by adding noise to latent features may be viewed as an approximation.

**[GR2] A comparison with previous SoTA**

We have compiled results from relevant papers and conducted additional experiments to provide a comprehensive comparison. Please note that hyperparameters and architectures may vary across studies. To account for this, we have added the results of our CCL approach with an architecture similar to the one used in PEPITA.

| Algorithm | CIFAR10 | CIFAR100 |
|--|--|--|
|FFA [1]| 59 (*) |-|
|CFSE [3]|78.11|51.23|
|SoftHebb [4]|80.30| 56.00|
|CCL (original)| 82.94 |56.29|
|PEPITA [2]|56.33|27.56|
|CCL (matched with PEPITA)|62.19|34.89|

Key observations:
- CCL outperforms other methods on both CIFAR10 and CIFAR100. Even with the simplified architecture used by PEPITA, CCL yields higher accuracy.
- SoftHebb [4] performs extensive layer-wise hyperparameter search (24 hyperparams), while CCL searches for only 3 parameters.

We will revise our paper to include this comprehensive comparison, providing a clearer context for our algorithm's performance relative to other recent work in the field.

**[GR3] Result clarification**

We thank the reviewers for carefully reading the results. We show the training loss during training in Figure (1.a) in the rebuttal document. BP actually optimizes the training objective better and yields much lower training loss than CCL but a high testing error, indicating that BP may be overfitting. We believe that more aggressive model regularization techniques may be helpful.

**[GR 4] Suggestion on reporting weight alignment**

We appreciate this insightful suggestion from reviewer sTDo and show the weight alignment between the forward and backward network in Figure (1.b) in the rebuttal document. Cosine similarity between the forward layer weights and the corresponding feedback layer weights are computed for the model trained on MNIST using a five-layered MLP architecture. Our findings reveal two distinct phases during training:
- Phase one: Layers 1 and 5 show rapid alignment increases.
- Phase two: Alignments in layers 1 and 5 saturate. Alignments in the intermediate layers gradually increase until saturation.

Interestingly, intermediate layers showed a bottom-up convergence pattern, with layer 2 achieving highest similarity first, followed by layers 3 and 4. We found that high alignment isn't always achieved or necessary for effective learning. This may be due to (1) Dimensional reduction: Information propagation through the network affects alignment, and (2) Non-linearity: Activation functions impact the relationship between forward and backward weights. These factors may contribute to observed alignment patterns without requiring perfect symmetry between forward and backward weights

(*) uses CNN with local receptive fields

[1] Hinton, G. (2022). The forward-forward algorithm: Some preliminary investigations.

[2] Dellaferrera, G., Kreiman, G., and Kreiman, G. (2022). Error-driven Input Modulation: Solving the Credit Assignment Problem without a Backward Pass.

[3] Andreas Papachristodoulou, Christos Kyrkou, Stelios Timotheou, and Theocharis Theocharides. (2024). Convolutional channel-wise competitive learning for the forward-forward algorithm.

[4] Journé, A., Rodriguez, H. G., Guo, Q., & Moraitis, T. (2023). Hebbian deep learning without feedback.

---

### Decision · Program_Chairs · 2024-09-25

**Decision:**

Accept (poster)

**Comment:**

The article presents a new biologically-plausible learning network that addresses the weight transport problem, the nonlocal update problem, and the backward locking problems of backpropagation. A key idea is to use a feedback network with trained weights. Reviewers agree to the novelty of the ideas in the paper, and note strong experimental results.